# Unraveling the link between neuropathy target esterase NTE/SWS, lysosomal storage diseases, inflammation, abnormal fatty acid metabolism, and leaky brain barrier

**Mariana I Tsap[1], Andriy S Yatsenko[1], Jan Hegermann[2], Bibiana Beckmann[3], Dimitrios Tsikas[3], Halyna R Shcherbata[1,4]\***

[1]Institute of Cell Biochemistry, Hannover Medical School, Hannover, Germany; [2]Institute of Functional and Applied Anatomy, Research Core Unit Electron Microscopy, Hannover Medical School, Hannover, Germany; [3]Institute of Toxicology, Hannover Medical School, Hannover, Germany; [4]Mount Desert Island Biological Laboratory, Bar Harbor, United States

**\*For correspondence:**
Shcherbata.Halyna@mh-hannover.de

**Competing interest:** The authors declare that no competing interests exist.

**Abstract** Mutations in *Drosophila* Swiss cheese (SWS) gene or its vertebrate orthologue neuropathy target esterase (NTE) lead to progressive neuronal degeneration in flies and humans. Despite its enzymatic function as a phospholipase is well established, the molecular mechanism responsible for maintaining nervous system integrity remains unclear. In this study, we found that NTE/SWS is present in surface glia that forms the blood-brain barrier (BBB) and that NTE/SWS is important to maintain its structure and permeability. Importantly, BBB glia-specific expression of *Drosophila NTE/SWS* or human NTE in the *sws* mutant background fully rescues surface glial organization and partially restores BBB integrity, suggesting a conserved function of NTE/SWS. Interestingly, *sws* mutant glia showed abnormal organization of plasma membrane domains and tight junction rafts accompanied by the accumulation of lipid droplets, lysosomes, and multilamellar bodies. Since the observed cellular phenotypes closely resemble the characteristics described in a group of metabolic disorders known as lysosomal storage diseases (LSDs), our data established a novel connection between NTE/SWS and these conditions. We found that mutants with defective BBB exhibit elevated levels of fatty acids, which are precursors of eicosanoids and are involved in the inflammatory response. Also, as a consequence of a permeable BBB, several innate immunity factors are upregulated in an age-dependent manner, while BBB glia-specific expression of NTE/SWS normalizes inflammatory response. Treatment with anti-inflammatory agents prevents the abnormal architecture of the BBB, suggesting that inflammation contributes to the maintenance of a healthy brain barrier. Considering the link between a malfunctioning BBB and various neurodegenerative diseases, gaining a deeper understanding of the molecular mechanisms causing inflammation due to a defective BBB could help to promote the use of anti-inflammatory therapies for age-related neurodegeneration.

## Editor's evaluation

The study underscores the essential role of Neuropathy Target Esterase (NTE)/Swiss Cheese (SWS) in preserving the blood-brain barrier (BBB) integrity and links its dysfunction to symptoms akin to lysosomal storage diseases, elevated fatty acid levels leading to abnormal cellular architecture, and

inflammation. It further elaborates on how a compromised BBB facilitates an inflammatory response and fatty acid accumulation, exacerbating neurodegenerative conditions. These important findings backed by solid evidence suggest that targeting inflammation and fatty acid dysregulation may offer therapeutic strategies for age-related neurodegeneration.

## Introduction

Aging is the major risk factor for neurodegenerative conditions, a group of disorders characterized by the progressive degeneration and dysfunction of the nervous system, which includes Alzheimer's and Parkinson's disease, amyotrophic lateral sclerosis, frontotemporal dementia, and many others. These diseases typically result in the gradual loss of cognitive function, movement control, and other neurological functions. The exact causes of neurodegenerative diseases are often complex and not fully understood, but they can involve a combination of genetic, environmental, and lifestyle factors.

Growing evidence suggests that inflammation plays a crucial role in age-related neurodegenerative diseases (*Zuo et al., 2019*; *Liu et al., 2018*; *Rojas-Gutierrez et al., 2017*; *Liu et al., 2017*; *Zhang et al., 2023*). Older organisms frequently develop chronic, low-grade inflammation, a condition often named inflammaging, which is characterized by a sustained increase in inflammatory markers without apparent infection or injury (*Chitnis and Weiner, 2017*; *Franceschi et al., 2018*; *McGeer and McGeer, 2004*; *Li et al., 2023*). This phenomenon presents a potential target for anti-inflammatory therapy in neurodegenerative disorders. Strategies involving modulation of inflammatory signaling pathways have shown promise in both animal models and clinical trials, offering hopeful prospects for neurodegenerative disease therapy (*Zhang et al., 2023*). While research aims to identify therapeutic targets to alleviate the impact of inflammaging on neurological health, a more in-depth understanding of the molecular mechanisms underlying inflammaging is needed.

One feature associated with neuroinflammatory degenerative diseases is dysfunction of the blood-brain barrier (BBB) (*Takata et al., 2021*). Disruption of the BBB has been observed in patients with numerous neurodegenerative diseases (*Sweeney et al., 2018*; *Spencer et al., 2018*; *Munji et al., 2019*; *Blyth et al., 2009*; *Gray and Woulfe, 2015*; *Zhou et al., 2023*; *Whitson et al., 2022*). Since the BBB plays a crucial role in maintaining the homeostasis of the brain environment, its disruption allows the infiltration of immune cells and molecules that can trigger and sustain inflammatory responses within the brain (*Segarra et al., 2021*).

Furthermore, dysfunction in lysosomal pathways also has been implicated in Alzheimer's and Parkinson's disease and many other neurodegenerative disorders (*Issa et al., 2018*). The lysosome-endosomal system is tightly associated with the maintenance of cell homeostasis and viability, regulation of cell death, oncogenesis, autophagy, and inflammation (*Peng et al., 2019*). In particular, lysosomes are cellular organelles responsible for degrading cellular waste and maintaining cellular health. Dysfunction of lysosomal processes can lead to the accumulation of damaged cellular components and trigger inflammatory responses, contributing to the overall inflammaging phenomenon (*Peng et al., 2019*; *Aman et al., 2021*).

Fatty acid metabolism is another aspect linked to inflammaging (*Calder, 2020*; *Chew et al., 2020*; *Emre et al., 2021*). Changes in lipid composition and metabolism, particularly an increase in pro-inflammatory fatty acids, have been observed in inflammaging. These alterations can contribute to the perpetuation of inflammatory signaling and potentially impact neurodegenerative conditions (*Dumas et al., 2023*; *Freitas et al., 2017*). Thus, understanding the interplay between the BBB, lysosomes, fatty acid metabolism and inflammaging is crucial for unraveling the intricate mechanisms involved in age-related neurodegenerative diseases.

In addition, human age-related neurodegenerative diseases can be accelerated by different stresses, which include a wide array of factors such as infection, trauma, diet, or exposure to toxic substances. Interestingly, abnormalities in the human neuropathy target esterase (NTE), encoded by PNPLA6 (patatin-like phospholipase domain containing 6), gene are linked to both neurodegeneration types: toxin-induced and hereditary. NTE is a transmembrane protein anchored to the cytoplasmic face of the endoplasmic reticulum and acts as a phospholipase that regulates lipid membrane homeostasis (*Glynn, 2005*; *Read et al., 2009*; *Lush et al., 1998*). Continuous inhibition of NTE activity by the organophosphorus compound tri-ortho-cresyl phosphate causes axonal degeneration in the central nervous system (CNS) and peripheral nervous system (PNS), a neuropathy that was consequently

named organophosphate-induced delayed neuropathy, OPIDN (*Richardson et al., 2013*; *Richardson et al., 2020*). Moreover, mutations in the NTE gene cause Gordon-Holmes or Boucher-Neuhäuser syndromes (*Deik et al., 2014*; *Synofzik et al., 2014*; *Synofzik et al., 2015*; *Topaloglu et al., 2014*) and a motor neuron disease called hereditary spastic paraplegia type 39 (HSP 39), in which distal parts of long spinal axons degenerate, leading to limb weakness and paralysis (*McFerrin et al., 2017*; *Rainier et al., 2008*). Genetically, HSP classification is based on the genes of origin called spastic para-plegia genes, which is a large group (>80) of genes (*Fereshtehnejad et al., 2023*). Over the past few years, research has shown that HSP is associated with endo-lysosomal system abnormalities (*Allison et al., 2017*; *Lim et al., 2015*; *Namekawa et al., 2007*; *Renvoisé et al., 2014*; *Chang et al., 2014*).

Human studies play a crucial role in understanding the real-world impact of aging and neuro-degeneration. However, for various reasons like a long lifespan, ethical considerations, heteroge-neity, cohort effects, limited controls, etc., humans may not always be ideal subjects for age-related research. To address these challenges, human studies are often complemented by research in model organisms, providing a comprehensive perspective on aging mechanisms and interventions. In partic-ular, modeling human neurodegenerative diseases in various model organisms can provide us with needed knowledge about the first hallmarks of neurodegeneration and also signaling mechanisms that are disrupted upon aging. It was shown that NTE is widely expressed in the mouse brain, and its activity is essential for lipid homeostasis in the nervous system (*Glynn et al., 1998*; *Moser et al., 2000*). NTE deficiency results in the distal degeneration of the longest spinal axons, accompanied by swelling that encompasses accumulated axoplasmic material (*Read et al., 2009*). Specific deletion of NTE in the neuronal tissue induces neurodegeneration (*Akassoglou et al., 2004*). Despite its known molecular function, the mechanism by which it maintains nervous system integrity during hereditary and toxin-induced neurodegeneration remains unknown. *Drosophila melanogaster* is an excellent genetic model organism to investigate the molecular mechanisms of age-dependent neurodegen-erative diseases, and it has been widely used to identify potential drug targets against neurodegen-erative diseases (*Ma et al., 2022*; *Kretzschmar, 2022*). Moreover, the fly nervous system is a great system to shed light on the evolutionarily conserved signaling pathways underlying disease pathology. In *Drosophila*, more than 70% of genes related to human diseased are conserved (*Ugur et al., 2016*). Studying human disease-related genes in *Drosophila* avoids the ethical issues of biomedical research involving human subjects.

Moreover, *Drosophila* serves as a well-defined model to study immune reactivity. Flies exhibit a robust immune response to septic injury, involving hemocytes (macrophage-like cells) that efficiently clear pathogens through phagocytosis. This involves the recruitment of immune cells and the activa-tion of immune-related genes. For instance, the signaling cascade of the glial cells missing transcrip-tion factor, which governs immune cell development and is triggered by aging and acute challenges, is conserved from flies to humans (*Pavlidaki et al., 2022*). Additionally, the immune response includes antimicrobial peptides (AMPs) secreted by fat body cells, activated by Toll and immune deficiency (IMD) pathways. The IMD pathway, triggered by Gram-negative bacteria, facilitates macrophage inva-sion into the inflamed brain, mediated by glia cells (*De Gregorio et al., 2002*). Macrophages in the brain can phagocytose synaptic material, impacting locomotor abilities and longevity, highlighting the delicate balance in evolutionary inflammatory responses (*Winkler et al., 2021*). Together, *Drosophila* satisfies many of the requirements to study human diseases that allows scientists, not only dissection on cellular and molecular levels but also investigation of behavior and neurodegeneration during aging (*Carney et al., 2023*; *Yatsenko and Shcherbata, 2021*; *Yatsenko et al., 2021*). Considering the increasing evidence linking inflammation and neurodegeneration in humans, gaining insights into the interplay between neuroinflammation and neurodegenerative processes in the *Drosophila* brain should be beneficial.

In this study, we used a *Drosophila* NTE/SWS model for human neurodegeneration. Swiss cheese protein (NTE/SWS) is a highly conserved lysophospholipase that can regulate phosphatidylcholine metabolism (*Lush et al., 1998*; *Zaccheo et al., 2004*). It was also shown that NTE/SWS can act as a regulator of the PKA-C3 catalytic subunit of protein kinase A (*Bettencourt da Cruz et al., 2008*; *Wentzell et al., 2014*). Loss of *Drosophila* NTE/SWS and vertebrate NTE has been shown to result in lipid droplet accumulation, which is involved in neurodegeneration pathogenesis (*Chang et al., 2019*; *Farmer et al., 2020*; *Melentev et al., 2021*). Loss of *sws* leads to age-dependent neuro-degeneration (*Figure 1B*, arrows), CNS vacuolization, and abnormal glial morphology accompanied

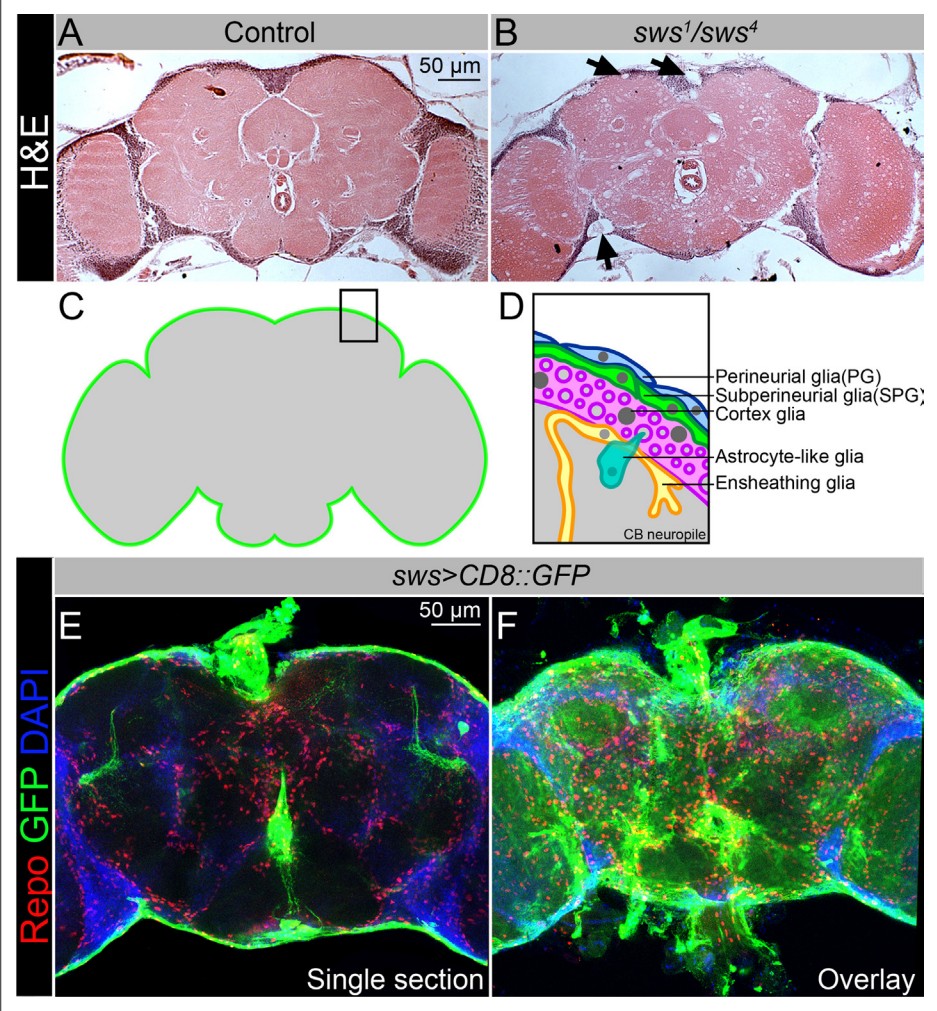

**Figure 1.** NTE/SWS is expressed in *Drosophila* brain and its loss leads to severe neurodegeneration. (**A–B**) Hematoxylin and eosin (H&E)-stained paraffin-embedded brain sections of the 30-day-old control (*OregonR x white^1118*, **A**) and 30-day-old *sws^1/sws^4* transheterozygous flies (**B**). Arrows indicate neurodegeneration at the brain surface. Scale bar: 50 μm. (**C–D**) Schemes of glia organization in the adult *Drosophila* brain – perineurial glia (PG, blue), subperineurial glia (SPG, light green), cortex glia (pink), astrocyte-like glia (turquoise), and ensheathing glia (yellow). (**E–F**) Expression pattern of *sws-Gal4* determined by combining of the transcriptional activator Gal4 under control of the *sws* gene promotor (*sws-Gal4*) and the *UAS-CD8::GFP* construct. Fluorescence images of the brain show that *sws* is expressed in all brain cells and strongly expressed in the surface glia (**E** – single section, **F** – Z-stack maximum projection). For NTE/SWS antibody staining pattern, see *Figure 2—figure supplement 1A and B*. Glia cells are marked with Repo (red), *sws* expression is marked by the membrane *CD8::GFP* (green), and nuclei are marked with DAPI (blue). Scale bar: 50 μm.

by the formation of multilayered glial structures in the adult *Drosophila* brain (*Kretzschmar et al., 1997*; *Dutta et al., 2016*). Recent studies have shown that the pan-glial knockdown of *sws* leads to increased levels of reactive oxygen species (ROS), which in turn induces oxidative stress (*Ryabova et al., 2021*). However, the role of NTE/SWS in distinct glial types is not clearly understood.

Similar to multiple other organisms, the *Drosophila* nervous system is composed of neurons and glial cells. Commonly recognized nomenclature identifies six distinct glial cell types based on morphology and function: perineurial (PG) and subperineurial glia (SPG), cortex glia, astrocyte-like and ensheathing glia, and finally the PNS-specific wrapping glial cells (*Yildirim et al., 2019*; *Trébuchet et al., 2019*). All organisms with a complex nervous system developed BBB to isolate their neurons from blood (*Limmer et al., 2014*). In higher vertebrates, this diffusion barrier is established by polarized endothelial cells that form extensive tight junctions (*Armulik et al., 2010*), whereas in lower

vertebrates and invertebrates the BBB is entirely formed by glial cells, which are additionally sealed by septate junctions (SJs) (*Limmer et al., 2014*). The *Drosophila* BBB includes two glial cell layers: the PG cells are primarily involved in nutrient uptake, whereas the main diffusion barrier is made by the SPG, which form pleated SJs (*Babatz et al., 2018*; *Schwabe et al., 2017*; *Kremer et al., 2017*). Glial cells in the *Drosophila* BBB play a crucial role in the immune response as they contribute to the maintenance of the BBB and respond to immune challenges (*Winkler et al., 2021*; *van Alphen et al., 2022*; *Kounatidis and Chtarbanova, 2018*; *Shu et al., 2023*).

Here, we showed that NTE/SWS is present in the surface glia of *Drosophila* brain that form the BBB and that NTE/SWS is important for the integrity and permeability of the barrier. Importantly, glia-specific expression of *Drosophila NTE/SWS* or human NTE in the *sws* mutant background fully rescues surface glial organization and partially restores BBB integrity, suggesting a conserved function of NTE/SWS. An important observation upon *sws* deficit was the formation of intracellular accumulations within lysosomes, which is a characteristic feature of lysosomal storage disorders (LSDs). Additionally, NTE/SWS regulates lipid metabolism, distribution of cell junction proteins, and organization of membrane rafts in BBB glia. Moreover, our research revealed that mutants with defective BBB exhibit elevated levels of several innate immunity factors as well as free fatty acids (FFAs), which are known to play a role in inflammatory pathways. Importantly, the BBB phenotype can be alleviated by the administration of anti-inflammatory agents. These findings emphasize the complex interplay between NTE/SWS, BBB function, inflammation, and innate immunity, providing potential avenues for therapeutic interventions in related disorders.

## Results
### SWS is expressed in the surface glia of *Drosophila* brain

Our previous data showed that NTE/SWS function is important for both glia and neuronal cells in the brain (*Melentev et al., 2021*; *Ryabova et al., 2021*). After downregulation of NTE/SWS in neurons, adult flies show a decrease in longevity, locomotor and memory deficits, and severe progression of neurodegeneration in the brain (*Melentev et al., 2021*). We have shown that NTE/SWS plays a role in the development of the learning center of the brain involved in short-term and long-term memory storage, olfactory control, and startle-induced locomotion (*Melentev et al., 2021*). In addition, we found that flies with NTE/SWS deficiency in neurons or glia show mitochondrial abnormalities as well as accumulation of ROS and lipid droplets (*Melentev et al., 2021*; *Ryabova et al., 2021*). Now we have decided to determine the cell type in which NTE/SWS plays a determining role in the maintenance of brain health.

Similar to its human counterpart, NTE, which is found in virtually all tissues, including the nervous system (https://www.proteinatlas.org/ENSG00000032444-PNPLA6/tissue), NTE/SWS is ubiquitously expressed in *Drosophila* brain, detected by immunohistochemical analysis using SWS-specific antibodies (*Figure 2—figure supplement 1A*). NTE/SWS is a transmembrane phospholipase anchored to the cytoplasmic side of the endoplasmic reticulum to regulate lipid membrane homeostasis. Its cytoplasmic localization makes it difficult to determine precisely in which brain cell type it has more pronounced expression, as neurons and glia have very complex shapes and forms. Therefore, additional markers must be used to discriminate NTE/SWS expression in the brain. To address this, we expressed membrane-bound GFP under control of the *sws* promoter (*sws-Gal4; UAS-CD8::GFP*), which allows labeling of membranes of cells in which the *sws* promoter is active. Importantly, *sws* was strongly expressed in the glia that surround the brain and form the blood-brain selective permeability barrier (*Figure 1E and F*). In *Drosophila*, the BBB is entirely made by two glial cell layers: PG and SPG (*Figure 1C and D*). With the help of sophisticated SJs, SPG cells form a tight barrier that prevents paracellular diffusion and separates the CNS from hemolymph. Since the BBB is protecting the brain from toxic substances, and NTE/SWS deregulation is associated with toxicity-induced neurodegeneration, we investigated whether NTE/SWS has a functional role in BBB maintenance and its selective permeability.

## Downregulation of NTE/SWS cell-autonomously affects surface glia integrity

To test if loss of NTE/SWS affects the barrier structure, we analyzed the expression pattern of Coracle (CoraC), which is a major component of SJs (*Yi et al., 2008*). In controls, CoraC is strongly expressed by SPG cells, shown as a smooth line at the brain surface (*Figure 2A*, green arrow). Upon *sws* loss, the CoraC pattern at the brain surface was broken and contained lesions and membrane aggregations (*Figure 2B*, magenta arrow).

Previous characterization of the *sws* loss-of-function mutant showed that NTE/SWS deficiency resulted in the formation of membranous glial structures, especially in the lamina cortex (*Kretzschmar et al., 1997*). Since NTE/SWS is ubiquitously expressed, we utilized the double driver line (*repo, nSyb-Gal4*, *Figure 2—figure supplement 1F and F'*) to achieve its downregulation in both neuronal and glial cells (*Figure 2—figure supplement 1C*). Since these animals had the same disorganized structure of brain surface as the loss-of-function mutant, we concluded that NTE/SWS functions specifically in the nervous system to preserve brain surface structure. Moreover, downregulation of *sws* in all glial cells (*repo>sws^RNAi*) resulted in the same phenotype (*Figure 2—figure supplement 2C*). At the same time, upon *sws* downregulation in neurons, we did not observe formation of lesions and membrane clusters in the brain surface (*Figure 2—figure supplement 2E*), indicating a cell-autonomous function of NTE/SWS in glia to maintain BBB organization.

To test if NTE/SWS has a cell-autonomous role in the brain barrier cells, we used already existing SPG driver lines (*moody-Gal4, UAS-CD8::GFP* and *Gli-Gal4, UAS-CD8::GFP*, *Figure 2—figure supplement 1D and E*) and *UAS-sws^RNAi*. We found that, similar to pan-glial *sws* knockdown, its downregulation specifically in SPG cells caused the formation of lesions and membrane clusters within the brain surface (*Figure 2C*, *Figure 2—figure supplement 2D*, blue and magenta arrows). Importantly, expression of *Drosophila* or human NTE in these glia cells rescued this phenotype (*Figure 2D and H*, *Figure 2—figure supplement 2F*), demonstrating the conserved function of this protein in SPG cells for brain surface formation and possibly maintenance of the brain barrier.

There have been remarkable recent advancements in the field of protein structure prediction, offering valuable tools for exploring three-dimensional structures with unprecedented effectiveness. We used the AlphaFold2 prediction and the PyMol tools (*Jumper et al., 2021*) to display predicted structure models of the human NTE and *Drosophila* NTE/SWS proteins. Both proteins contain a highly conserved patatin-like phospholipase (EST) domain (*Figure 2—figure supplement 3*, EST domain in magenta). EST domains in NTE/SWS (952–1118) and human NTE (981–1147) demonstrated a remarkably high level of confidence, exhibiting helical structures with predicted local distance difference test scores (pLDDT) exceeding 90 (*Figure 2—figure supplement 3*).The EST domain exhibits a distinctive architectural pattern comprising three layers of α/β/α structure. Its central region is formed by a six-stranded β-sheet, flanked by α-helices in the front and back. Upon comparing the predicted structures of EST-SWS and EST-NTE, we observed a significant overlap between them (*Figure 2—figure supplement 3*). These findings offer additional evidence of the high conservation of functional domains in NTE/SWS and the close relationship between these proteins across different species.

Together, the remarkable similarities observed between human and *Drosophila* SWE/NTE protein structure along with their shared involvement in the formation and maintenance of the brain barrier in *Drosophila* emphasize their close relationship and suggest a conserved function in BBB maintenance.

## Downregulation of NTE/SWS results in multilamellar accumulations

Next, we aimed to understand the nature of the SPG phenotype caused by *sws* deficiency. SPG cells have a very specific shape; they are thin and very large. Fewer than 50 SPG cells surround one adult brain hemisphere and a single SPG cell can cover the size of one half of the imaginal disc of the eye, covering an area equivalent to approximately 10,000 epithelial cells (*Limmer et al., 2014*; *Hartenstein, 2011*; *Silies et al., 2007*). Therefore, to better visualize the defects in surface glia organization upon *sws* loss, we introduced *moody-Gal4, UAS-CD8::GFP* (*moody>CD8::GFP*) constructs into the *sws^1* mutant background, which allowed analysis of SPG cell membranes. To our surprise, we observed that almost all lesions that were formed near the brain surface contained membrane material marked by *CD8::GFP* (*Figure 2F*). This was in sharp contrast to the control, where SPG membranes formed a distinct GFP-positive line (*Figure 2E*). Importantly, the same excessive SPG cell membranes were observed inside the lesions formed upon *sws* downregulation explicitly in SPG cells (*Figure 2G*),

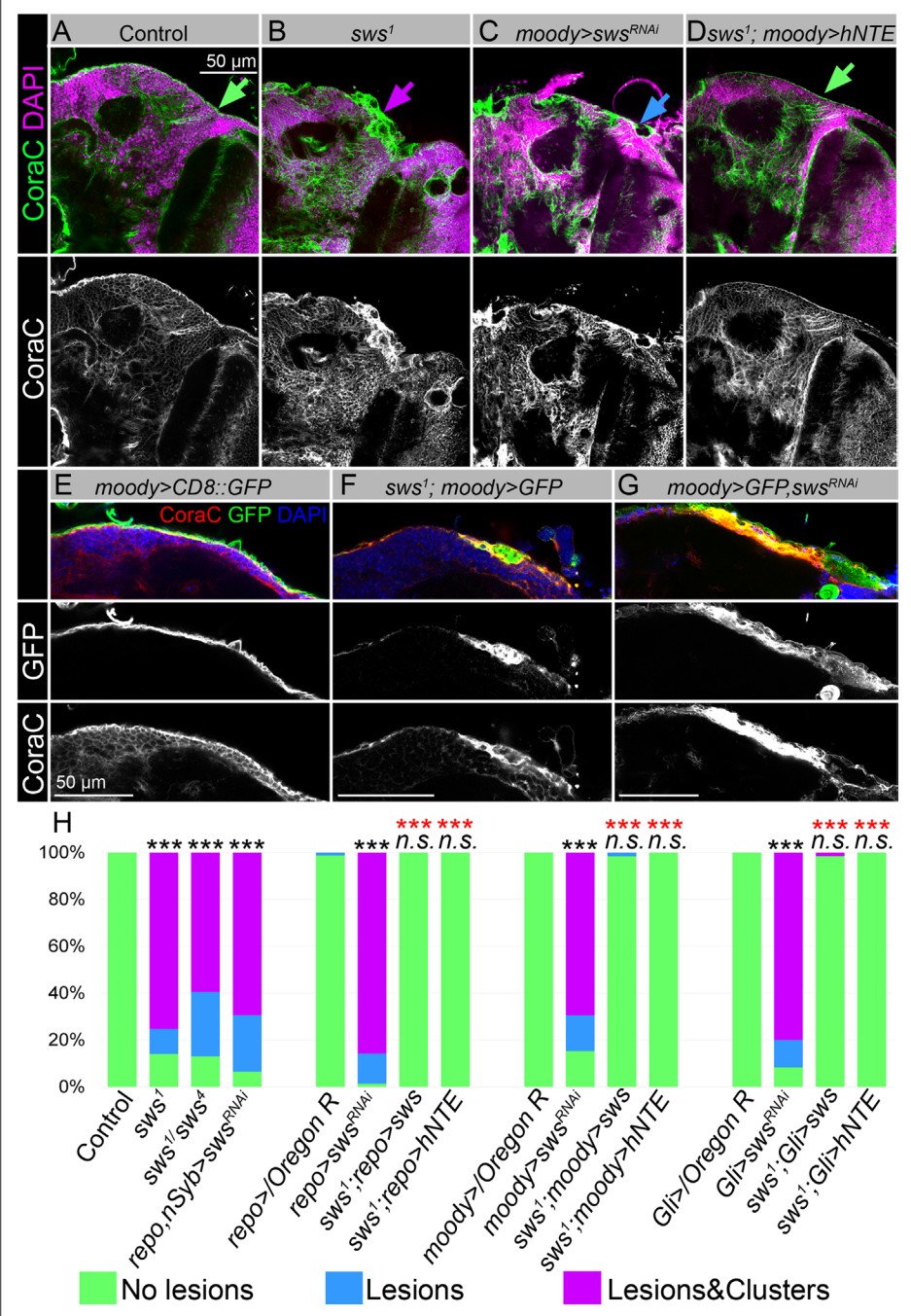

**Figure 2.** Downregulation of NTE/SWS affects surface glia architecture. (**A–D**) Adult brains stained with Coracle (CoraC) (green) and DAPI (magenta). (**A**) In controls (*Oregon R x white*[1118]), CoraC expression is pictured as the smooth line at the surface of the brain (green arrow). In *sws*[1] mutants (**B**) and in mutants with *sws* downregulation in subperineurial glia (SPG) cells (*moody>sws*[RNAi], **C**) the outer glial cell layer labeled by CoraC is irregular and contains either lesions (blue arrow) or lesions and membrane clusters (magenta arrow). Expression of human NTE (*sws*[1]; *moody>hNTE*, **D**) in SPG cells in mutant background results in the brain surface appearance which is similar to control (green arrow). Scale bar: 50 µm. (**E–G**) Adult brains stained with CoraC (red), GFP (green), and DAPI (blue) to detect SPG cell membranes marked by co-expression of CoraC and *moody>CD8::GFP* (red + green = yellow). (**E**) A smooth line of SPG cell membranes is observed at the surface of control brains (*moody>CD8::GFP*). (**F**) In *sws* loss-of-function mutants (*sws*[1]; *moody>CD8::GFP*), most of the vacuoles formed near the brain surface are filled with the GFP-positive SPG membranes. (**G**) Downregulation of *sws* specifically in SPG cells (*moody>sws*[RNAi]) results in the appearance of the same excessive SPG cell membranes inside the brain lesions.

*Figure 2 continued on next page*

*Figure 2 continued*

Scale bar: 50 µm. (**H**) Bar graph shows the percentage of the brain hemispheres with defective brain surface. The percentage of the brain hemispheres with normal brain surface is shown in green, the percentage of the brain hemispheres containing lesions is shown in blue, and the percentage of the brain hemispheres with formed lesions and membrane clusters within the brain surface is shown in purple. Two-way tables and chi-squared test were used for statistical analysis. *p<0.05, **p<0.005, ***p<0.001, black asterisks – compared to *Gal4-driver x OR*, red asterisks – compared to *Gal4-driver x UAS-sws^{RNAi}*, number of adult brain hemispheres ≥43, at least three biological replicates (see **Supplementary file 2**).

The online version of this article includes the following figure supplement(s) for figure 2:

**Figure supplement 1.** NTE/SWS expression pattern, *sws* mRNA levels, and expression patterns of the Gal4 driver lines used in the study.

**Figure supplement 2.** *sws* downregulation in neurons does not result in the formation of lesions and membrane clusters within the brain surface, and expression of *Drosophila* NTE/SWS in glia cells rescued glial phenotype.

**Figure supplement 3.** 3D structures of human NTE and *Drosophila* SWS.

---

confirming that NTE/SWS is required cell-autonomously in SPG cells for the proper architecture of the surface glia.

Next, we wanted to understand the origin of these excessive membranes observed in *sws*-deficient glial cells. This task appeared to be quite challenging, as SPG cells form a very thin polarized endothelium, not even reaching 1 µm thickness in most areas (*Limmer et al., 2014*). In addition, SPG cells localize in very close proximity to each other and to neurons, making the analysis of subcellular protein localization challenging. Therefore, to dissect in more detail the *sws*-related phenotype of accumulated SPG membranes inside the lesions on the brain surface, we used an electron microscopy approach.

We found that *sws* mutants showed the formation of various multilamellar bodies in the brain, which were not observed in the control (*Figure 3A–B*). These atypical structures ranged in size from 5 to 15 µm and contained concentrically laminated and multilayered membranes (yellow arrows), lipid droplets (red arrows), and other partially degraded organelles or cytoplasmic constituents. We hypothesized that these inclusions most likely correspond to secondary lysosomes in the phase of digesting endosomal cargo, which are a hallmark of lysosomal storage diseases (LSDs).

To authenticate the nature of membranous accumulation in *sws* mutants, we used endosomal and lysosomal markers. Rab7 is a small GTPase that belongs to the Rab family and controls transport to late endocytic compartments such as late endosomes and lysosomes (*Guerra and Bucci, 2016*). Immunohistochemical analysis demonstrated that in contrast to controls, where Rab7 was present in relatively small and evenly dispersed throughout the brain late endosomes and lysosomes (*Figure 3D*, red), in *sws*-deficient brains, accumulation of Rab7-positive compartments was observed. Moreover, Rab7-positive structures colocalized with atypical membrane aggregates of SPG cells (*Figure 3D'*, yellow). The same assemblies were observed upon *sws* downregulation in SPG cells (*Figure 3D'*, yellow).

Rab7 controls biogenesis of lysosomes and clustering and fusion of late endosomes and lysosomes (*Feng et al., 2014*). Therefore, to support the idea that these abnormal cellular accumulations are of lysosomal origin, we used an additional marker – CathepsinL – which is a key lysosomal proteolytic enzyme expressed in most eukaryotic cells (*Xu et al., 2021*). We found that *sws* loss or its downregulation in barrier-forming glia cells resulted in the appearance of CathepsinL-positive inclusions that colocalized with GFP-labeled membrane aggregates formed in the mutant SPG cells (*Figure 3E–E''*, yellow). We conclude that the structures observed upon NTE/SWS deregulation are abnormally enlarged lysosomes.

Next, we quantified the number of brain hemispheres with atypical Rab7- or CathepsinL-positive accumulations. In the control groups, very few (<10%) of the analyzed brains showed accumulation of Rab7 or CathepsinL. However, in mutants with *sws* loss-of-function and with *sws* SPG-specific downregulation, a significant increase in the frequency of brains containing Rab7- or CathepsinL-positive aggregates was observed (*Figure 3F and G*). Since *sws*-associated neurodegeneration is age-dependent (*Melentev et al., 2021*; *Kretzschmar et al., 1997*; *Dutta et al., 2016*; *Sujkowski et al., 2015*; *Sunderhaus et al., 2019*), we tested if abnormal lysosomes positive for Rab7 and CathepsinL increase with age. Analysis of the brains of 15-day-old *sws* downregulation in SPG cells demonstrated ~2-fold increase in the percentage of brains with lysosomal accumulations within the

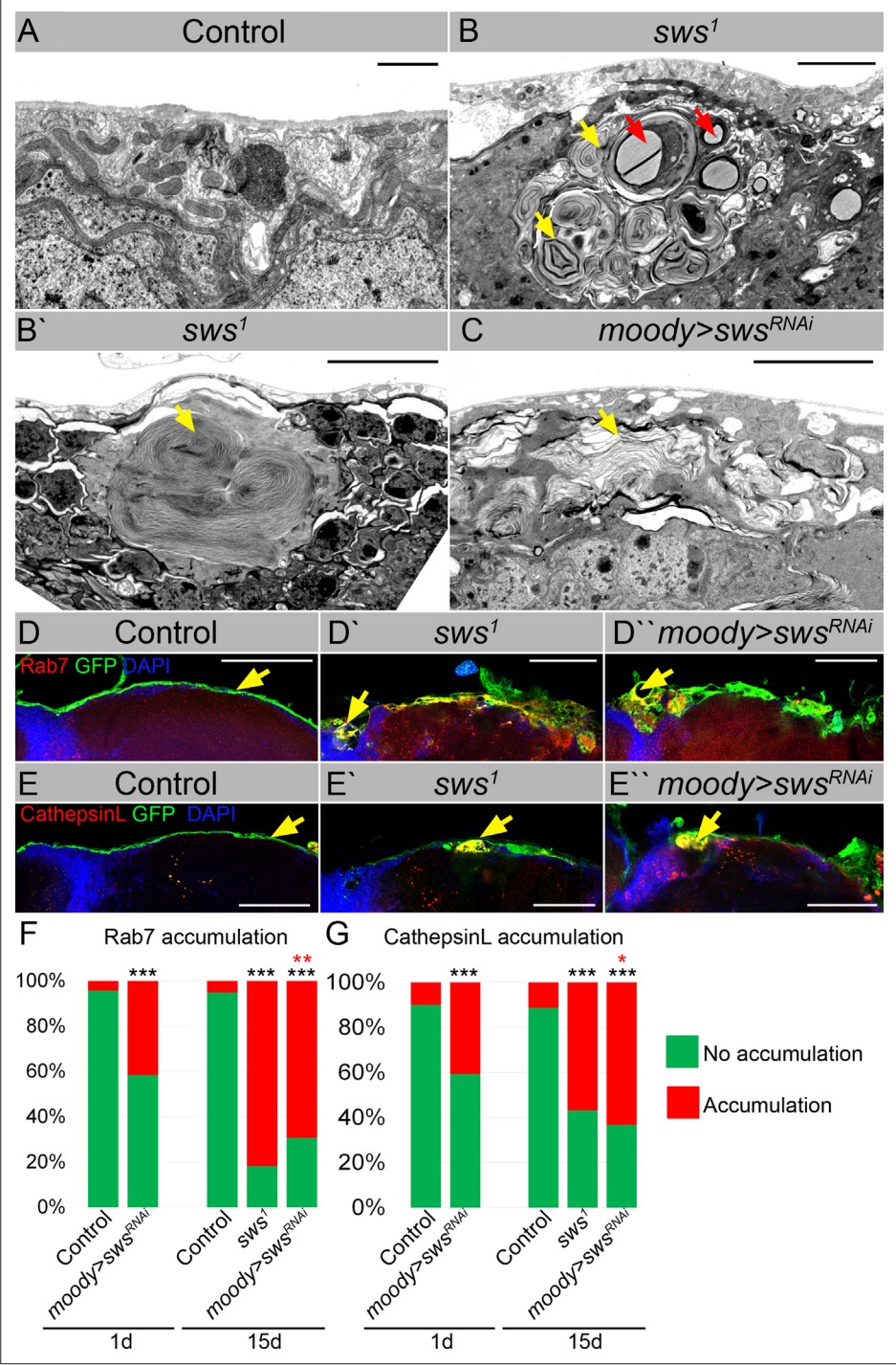

**Figure 3.** Downregulation of NTE/SWS results in intracellular accumulations. (**A–C**) Electron microscopy images of the surface area of the adult brains. (**A**) In controls (*white[1118]*), glia cells that do not contain any abnormal subcellular structures. Scale bar: 1 µm. (**B–B'**) *sws[1]* mutant brains have irregular surface and abnormal accumulation of endomembranous structures (yellow arrows) and lipid droplets (red arrows). Scale bar: 5 µm. (**C**) *moody>sws[RNAi]* fly brain has same abnormal accumulations of endomembranous structures (yellow arrows) as *sws* mutant. Scale bar: 5 µm. (**D–D'**) Adult brains stained with Rab7 (red) to detect lysosomes and late endosomes, GFP (*moody>CD8::GFP*, green) to mark subperineurial glia (SPG) cell membranes and DAPI (blue) to mark nuclei. (**D**) A smooth line of SPG cell membranes is observed at the surface of control brains (*moody>CD8::GFP*, green), Rab7 is present in relatively small amounts and evenly dispersed throughout in the brain (red). (**D'**) In *sws* loss-of-function mutants (*sws[1]; moody>CD8::GFP*), Rab7-positive structures colocalized with atypical membrane aggregates

*Figure 3 continued on next page*

*Figure 3 continued*

of GFP-positive SPG membranes (red + green = yellow). (**D'**) Downregulation of *sws* specifically in SPG cells (*moody>sws^RNAi*) results in the appearance of the same assemblies in the SPG cells (yellow). Scale bar: 50 µm. (**E–E'**) Adult brains stained with CathepsinL (red) to detect lysosomes, GFP (*moody>CD8::GFP*, green) to mark SPG cell membranes, and DAPI (blue) to mark nuclei. (**E**) A smooth line of SPG cell membranes is observed at the surface of control brains (*moody>CD8::GFP*, green), CathepsinL is present in relatively small amounts in the brain (red). (**D**) In *sws* loss-of-function mutants (*sws¹; moody>CD8::GFP*), CathepsinL-positive structures colocalized with atypical membrane aggregates of GFP-positive SPG membranes (red + green = yellow). (**E'**) Downregulation of *sws* specifically in SPG cells (*moody>sws^RNAi*) results in the appearance of the same assemblies in the SPG cells (yellow). Scale bar: 50 µm. (**F**) Bar graph shows the percentage of brains with accumulated Rab7 structures at the brain surface. Two-way tables and chi-squared test were used for statistical analysis. *$p<0.05$, **$p<0.005$, ***$p<0.001$, black asterisks – compared to *moody-Gal4 x OR*, red asterisks – compared to 1-day-old *moody-Gal4 x UAS-sws^RNAi*, number of adult brain hemispheres ≥44, at least three biological replicates (see **Supplementary file 3**). (**G**) Bar graph shows the percentage of brains with accumulated CathepsinL structures at the brain surface. Two-way tables and chi-squared test were used for statistical analysis. *$p<0.05$, **$p<0.005$, ***$p<0.001$, black asterisks – compared to *moody-Gal4 x OR*, red asterisks – compared to 1-day-old *moody-Gal4 x UAS-sws^RNAi*, number of adult brain hemispheres ≥49, at least three biological replicates (see **Supplementary file 3**).

---

brain surface in comparison to 1-day-old animals (*Figure 3F and G*). These data demonstrate for the first time that NTE/SWS-associated phenotypes might be additionally characterized by the excessive storage of cellular material in lysosomes that is accelerated by age.

Importantly, similar abnormal buildup of cellular material in lysosomes have been found in hippocampal neuropil (*Akassoglou et al., 2004*) and spinal axons of NTE-deficient mice (*Read et al., 2009*). While these structures have not been specifically described as lysosomal defects, the presence of similar dense bodies containing concentrically laminated and multilayered membranes in NTE-deficient mice suggests that, similar to *Drosophila*, NTE/SWS-related phenotypes in mammals may also be associated with excessive storage of cellular material in lysosomes. Lysosomal changes and dysfunction have been involved in the initiation and development of numerous diseases, such as cancer, autoimmune, cardiovascular, neurodegenerative, and LSDs (*Cao et al., 2021*; *Hebbar et al., 2017*). In particular, LSDs are a group of rare metabolic disorders caused by inherited defects in genes that encode proteins vital for lysosomal homeostasis, such as lysosomal hydrolases or membrane proteins. LSDs often manifest as neurodegenerative disorders. Therefore, next, we wanted to investigate how lysosomal accumulation in SPG cells affects their functions, resulting in progressive brain degeneration.

## Downregulation of NTE/SWS affects brain permeability barrier

The main function of SPG cells is to protect the CNS from being exposed to molecules that are harmless to peripheral organs but toxic to brain neurons. SPG cells form a thick polarized endothelium, selective permeability of which is achieved by forming very tight SJs that provide structural strength and a barrier that controls the flow of various solutes from outside the brain (*Limmer et al., 2014*; *Hartenstein, 2011*; *Silies et al., 2007*). Since our data show that the expression pattern of a key SJ protein, CoraC, is dramatically perturbed in *sws* mutant brains (*Figure 2A and B*), we decided to test if deregulation of NTE/SWS can affect the ability of SPG cells to form a selective permeability barrier.

As a result of abnormal BBB function, the CNS becomes permeable to small molecules such as dextran-coupled dyes. To test BBB permeability, the 10 kDa dextran dye was injected into the abdomen of flies (*Figure 4A*). After injection, animals were allowed to recover for at least 12 hr, followed by the dissection and analysis of adult brains. In controls, dextran dye predominantly remained at the outer surface of the brain (*Figure 4B and B'*). In contrast, the dye was detected inside almost all of the *sws¹* mutant brains (*Figure 4C and C'*). Moreover, the downregulation of *sws* in different types of glial cells also caused increased permeability of brain barrier in more than 80% of the analyzed brains (*Figure 4D*). Expression of *Drosophila* NTE/SWS and human NTE in glia in *sws¹* mutant rescued the organization of the surface glia (*Figure 2H*) and partially rescued the barrier phenotype, suggesting that human NTE and *Drosophila* NTE/SWS are important for the BBB integrity in *Drosophila* (*Figure 4D*). Taken together, our results demonstrate that SPG cells with NTE/SWS deficiency are characterized by defective brain barrier function and lysosomal accumulation of excess cellular material, which includes membranes.

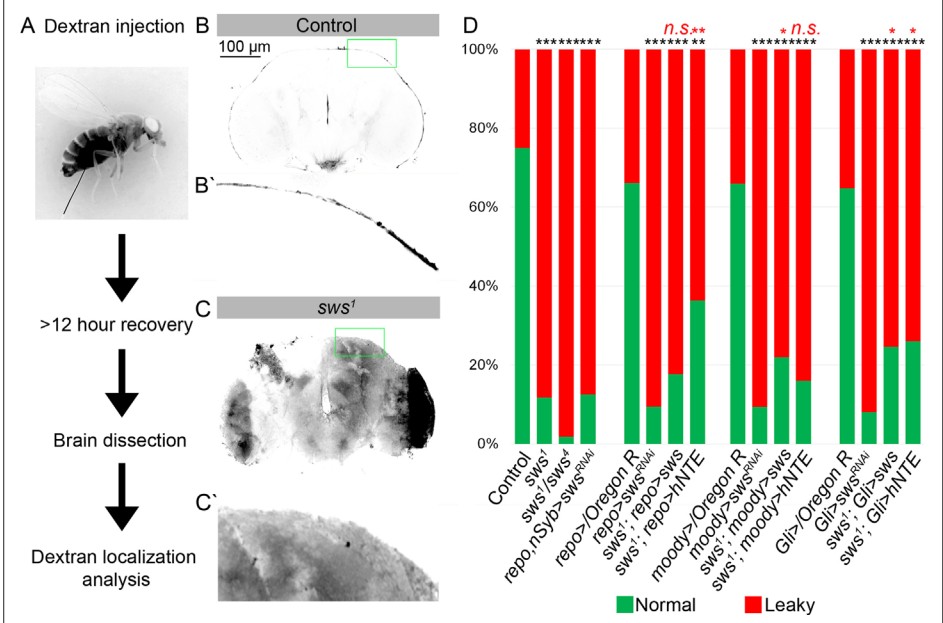

**Figure 4.** Downregulation of NTE/SWS affects brain permeability barrier. (**A**) Scheme of 10 kDa dextran dye permeability assay (see also Materials and methods for a detailed description of the procedure). (**B–C**) Localization of dextran dye more than 12 hr after injection in control (*Oregon R*) flies (**B–B'**) and in *sws¹* mutant (**C–C'**). Note that dextran dye can be detected in the cells present inside the mutant brain in contrast to control, where dye stays at the outer surface of the brain. Scale bar: 100 μm. (**D**) Bar graph shows the percentage of the brains with the defective permeability barrier. Two-way tables and chi-squared test were used for statistical analysis. *p<0.05, **p<0.005, ***p<0.001, black asterisks – compared to *Gal4-driver x OR*, red asterisks – compared to *Gal4-driver x UAS-sws^RNAi*, number of adult brain hemispheres ≥44, at least three biological replicates (see ***Supplementary file 4***).

Next, we wanted to understand whether the compromised brain barrier in *sws* mutants triggers the activation of any cellular stress pathways, including apoptosis, ferroptosis, oxidative stress, ER stress, and inflammation. We treated mutant flies for 14 days with different anti-inflammatory substances and stress suppressors and analyzed whether observed glial phenotypes could be suppressed by any medication. We analyzed CoraC expression and compared the frequencies of abnormal brain surface appearance in the drug-treated versus untreated animals (***Figure 5—figure supplement 1A, B***). We revealed that sodium salicylate, a non-steroidal anti-inflammatory drug (NSAID) and rapamycin, which activates autophagy by inhibiting Tor (***Xu et al., 2017***), showed the best ability to suppress surface glia phenotypes in *sws* mutants (***Figure 5—figure supplement 1B, C'***). This indicates that an activated inflammatory response is associated with *sws* deficit.

### *moody* flies with a permeable BBB show glial phenotype similar to *sws* mutants

A leaky BBB allows different toxic substances and bacteria to enter the CNS and affect neurons and glial cells, which can lead to cell death and increased inflammation in mammals (***Kim et al., 2012***). To test if a permeable brain barrier in general is causing inflammation in *Drosophila*, we decided to test if an additional mutant with defective BBB has an increased inflammatory response in the brain. We focused on a *moody^ΔC17* mutant that has been previously shown to have a defective brain barrier (***Bainton et al., 2005***).

First, we tested whether the *moody* mutant shows a phenotype similar to that observed in *sws* mutants by analysis of the CoraC expression pattern. We observed that the surface brain layer in *moody* mutants or upon *moody* downregulation in SPG by *moody-Gal4* (*moody>moody^RNAi*) contained lesions and had an abnormal membrane assembly, resembling CoraC expression pattern in *sws* mutants (***Figure 5—figure supplement 2A–C*** and ***Figure 2B***, magenta arrows).

Second, we analyzed if anti-inflammatory factors can reduce glial phenotypes in *moody* mutants, similar to *sws* mutants. We found that in *moody* mutants, the surface glia phenotype analyzed using CoraC as a marker could also be suppressed by NSAID and rapamycin (*Figure 5A*, *Figure 5—figure supplement 1D and D'*). The fact that anti-inflammatory factors can reduce glial phenotypes in both *sws* and *moody* mutants indicates that inflammation, triggered as a result of a compromised brain barrier, plays a role in a feedback loop that exacerbates the abnormal surface glia organization (*Figure 5G*). At the same time, inflammation inhibitors only partially rescued the BBB phenotype in *moody* and *sws* mutants, suggesting the involvement of additional pathways in maintaining the BBB.

## Mutants with defective BBB show upregulation of several innate immunity factors and FFAs

Next, we tested whether inflammatory pathways are activated in both mutants with permeable barriers. The molecular mechanisms of innate immunity between flies and mammals are highly evolutionarily conserved. For example, *Drosophila* Toll and IMD pathways are nuclear factor kappa B (NF-κB)-based signaling pathways that share similarities with the Toll-like receptor and tumor necrosis factor receptor 1 signaling pathways in mammals (*Pavlidaki et al., 2022*; *Kounatidis and Chtarbanova, 2018*; *Cao et al., 2013*; *Kounatidis et al., 2017*). It has been previously shown that in glial cells, activation of the IMD pathway results in phosphorylation of the NF-κB transcription factor Relish, which is translocated to the nucleus to induce expression of the AMPs Attacin A, Cecropin A, and Diptericin (*Winkler et al., 2021*; *Kounatidis and Chtarbanova, 2018*). We performed quantitative PCR (qPCR) analysis and measured the mRNA levels of these AMPs in heads of *sws* and *moody* loss-of-function mutants. We found that the mRNA levels of all three AMPs were significantly upregulated in mutants in comparison to relevant controls (*Figure 5B*). These data demonstrate that both mutants with defective BBB exhibit an increased inflammatory response.

In addition, polyunsaturated fatty acids (PUFAs) have been shown to play a key role in inflammatory processes. Their oxygenated products, called eicosanoids, induce and regulate inflammation via G-protein-coupled receptor (GPCR) signaling pathways (*Stanley and Kim, 2018*). To find out whether levels of polyunsaturated and saturated fatty acids are changed, we measured levels of FFAs from accurately weighed heads of control flies and mutants with defective BBB (*sws¹* and *moody^{ΔC17}*). FFAs were measured by gas chromatography-mass spectrometry (GC-MS) as described recently (*von Hanstein et al., 2023*). We found that both mutants with defective BBB show upregulated levels of linoleic acid, α- and γ-linolenic acid, eicosanoic acid, arachidonic acid, and eicosapentaenoic acid when compared to controls (*Figure 5C*). Additionally, levels of other FFAs involved in inflammatory response, 9-cis-tetradecenoic acid, palmitic acid, palmitoleic acid, stearic acid, and oleic acid (*Miao et al., 2015*; *Korbecki and Bajdak-Rusinek, 2019*) were elevated upon *sws* or *moody* loss (*Figure 5C*). These data show that in both mutants with a compromised BBB, the inflammatory response is accompanied by the accumulation of FFAs.

Given that the loss of *sws* results in age-dependent neurodegeneration (*Kretzschmar et al., 1997*), we investigated whether the increased inflammatory response is progressing with age. We performed qPCR analysis and quantified mRNA levels of AMPs (Attacin A, Cecropin A, and Diptericin) in the heads of *sws* mutants and flies that had *sws* downregulation only in SPG cells (*moody>sws^{RNAi}*) of 15- and 30-day-old flies. We confirmed that the mRNA levels of all three AMPs were significantly upregulated in mutants of both ages in comparison to the relevant controls (*Figure 5D*, black asterisks). Furthermore, we observed a significant age-related increase in the expression of inflammatory genes in both *sws* mutants and flies with *sws* downregulation in SPG cells (*moody>sws^{RNAi}*, *Figure 5D*, red asterisks), thereby illustrating the correlation between age-related NTE/SWS neurodegeneration and inflammatory processes. Importantly, expression of *Drosophila* NTE/SWS in SPG cells in *sws¹* mutant background normalized levels of inflammatory genes expression in flies of both ages (15- and 30-day-old flies), confirming that the increased inflammatory response is a consequence of the defective BBB (*Figure 5D*, green asterisks). Moreover, downregulating *sws* in glial cells during adulthood, after BBB formation, resulted in an increased inflammatory response (*Figure 5—figure supplement 1*). Since previous studies have demonstrated the induction of neurodegeneration by the overactivation of innate immune-response pathways, especially elevated expression of AMPs (*Cao et al., 2013*), our data showing increased levels of AMPs in aging flies with a defective BBB further strengthen the

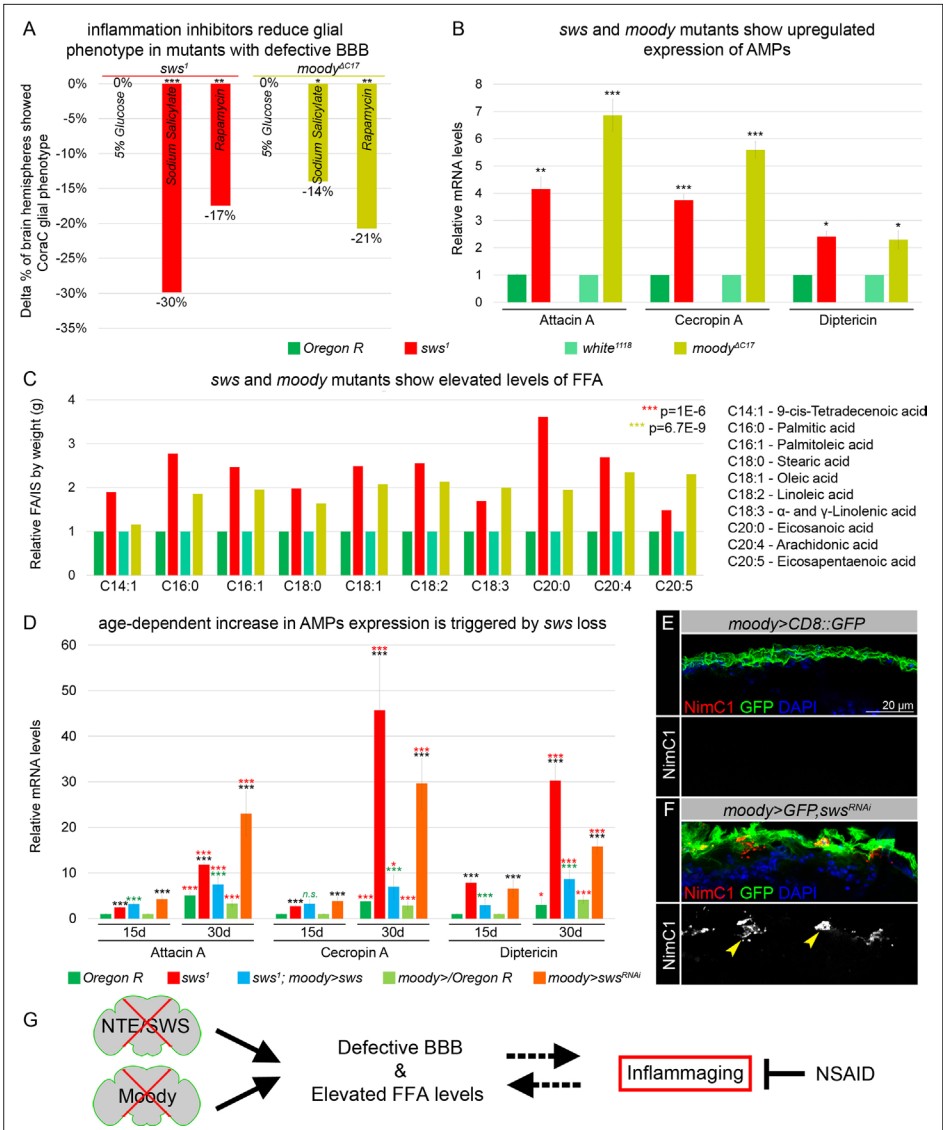

**Figure 5.** Mutants with defective blood-brain barrier (BBB) have an increased age-dependent inflammatory response and elevated levels of free fatty acid (FFA). (**A**) Bar graph shows the reduction in the percentage of the glial phenotype, assayed by Coracle (CoraC) expression pattern, in *sws[1]* (red) and *moody[ΔC17]* (olive) mutants that were treated with non-steroidal anti-inflammatory drug (NSAID) and rapamycin in comparison to untreated mutants.This suggests that inflammation accelerates surface glia phenotype. Two-way tables and chi-squared test were used for statistical analysis, number of adult brain hemispheres ≥104, p<0.05, **p<0.005, ***p<0.001, at least three biological replicates (see ***Supplementary file 5***). (**B**) Real-time quantitative PCR (RT-qPCR) analysis of antimicrobial peptides (AMPs) mRNA levels from relevant controls (green) and *sws[1]* (red) and *moody* (olive) mutant fly heads shows significantly upregulated expression of inflammatory response genes: *Attacin A, Cecropin A,* and *Diptericin*. AVE ± SEM is indicated. Two-tailed Student's test was used to test for statistical significance, *p<0.05, **p<0.005, ***p<0.001 (see ***Supplementary file 1***). (**C**) GS-MS measurements of free fatty acids (FFAs) indicate the relative increase of several FFAs in the heads of *sws[1]* (red) and *moody[ΔC17]* (olive) mutants compared to relevant controls (*Oregon R* and *white[1118]*, green). One-way ANOVA test was used for statistical analysis, *p<0.05, **p<0.005, ***p<0.001 (see ***Supplementary file 6a***). (**D**) RT-qPCR analysis of AMP mRNA levels from the heads of 15- and 30-day-old relevant controls (green), *sws[1]* (red), and *moody>sws[RNAi]* (orange) mutants shows the age-dependent upregulation of the expression of inflammatory response genes (*Attacin A, Cecropin A,* and *Diptericin*). Moreover, expression of *Drosophila* NTE/SWS (*sws[1]; moody>sws,* blue) in subperineurial glia (SPG) cells in mutant background normalizes levels of AMPs. The AVE ± SEM is shown. Two-tailed Student's test was used to test for statistical significance. p<0.05, **p<0.005, ***p<0.001 (see ***Supplementary file 1***). Black asterisks – *sws[1]* compared to *Oregon R; moody>sws[RNAi]* compared to *moody>/Oregon R* of the same age. Green asterisks – rescue, *sws[1];*

*Figure 5 continued on next page*

*Figure 5 continued*

*moody>sws* compared to *sws[1]*. Red asterisks – aging, 30-day-old compared to 15-day-old flies. (**E–F**) Adult brains stained with NimC1 (red), GFP (green), and DAPI (blue) to reveal the macrophage entry in the brain. Note that no macrophages marked by NimC1 (red) are detected in the control brain (*moody>CD8::GFP,* **E**), while NimC1-positive marcophages are detected in *moody>GFP, sws[RNAi]* brain (yellow arrowheads, **F**). Scale bar: 20 μm. (**G**) Mutants with defective brain barrier have upregulated innate immunity factors and exhibit elevated levels of FFAs involved in mediating the inflammatory response. Treatment with anti-inflammatory agents alleviates BBB phenotypes, suggesting that a signaling loop that links the condition of the brain barrier permeability, lipid metabolism, and inflammation.

The online version of this article includes the following source data and figure supplement(s) for figure 5:

**Source data 1.** GS-MS measurements of free fatty acids (FFA).

**Figure supplement 1.** *sws* mutants show increased inflammation and macrophage entry into the brains.

**Figure supplement 2.** *Moody* flies with a permeable blood-brain barrier (BBB) have similar to *sws* mutants brain surface appearance, but distinct septate junction phenotypes, and *moody[ΔC17]* mutant shows no accumulation of endosomal-lysosomal pathway components such as Rab7.

**Figure supplement 3.** Gas chromatography-mass spectrometry (GC-MS) analysis of free fatty acids (FFAs).

---

connection between the BBB, AMPs, and neuroinflammation and reinforce the causative link between BBB breakdown and inflammaging.

Upon infections and autoimmune conditions, macrophages have the capability to infiltrate the brain, aiding in pathogen removal but also posing the potential risk of causing tissue damage. It has been recently shown that the IMD pathway attracts and facilitates the invasion of hemolymph-borne macrophages across the BBB into the inflamed brain during pupal stages (*Winkler et al., 2021*). To investigate whether the neuroinflammatory response in *sws* mutants is associated with the entry of macrophages into the brain, we introduced *srp(Hemo)3xmCherry*, which enables the labeling of macrophages (*Cattenoz et al., 2021*), into the *sws[1]* mutant background. In contrast to control brains, we observed the presence of macrophages within the brain in both developing and adult brains of *sws[1]; srp(Hemo)3xmCherry* mutants (yellow arrowheads, *Figure 5—figure supplement 1F and G*). Moreover, using the anti-NimC1 antibody (*Kurucz et al., 2007*), macrophage infiltration into the adult brain was detected in flies with *sws* downregulation specifically in SPG cells (*moody>GFP, sws[RNAi]*, yellow arrowheads, *Figure 5E and F*). This suggests that the presence of an inflammatory response in mutants with a compromised BBB is associated with macrophage entry into the brain.

## *sws* and *moody* mutants have distinct surface glia phenotypes

However, while both *sws* and *moody* mutants have defective BBB, the nature of these mutations and their involvement in cellular processes are very different. Moody is a GPCR that is expressed in SPGs and localizes to the sites of SJ formation (*Babatz et al., 2018*; *Bainton et al., 2005*; *Schwabe et al., 2005*; *Li et al., 2021a*). Its cellular function is to control continued cell growth of SPG by differentially regulating actomyosin contractility and SJ organization (*Li et al., 2021a*). NTE/SWS is a transmembrane ER protein that hydrolyzes phosphatidylcholine and binds to and inhibits the C3 catalytic subunit of protein kinase A (*Bettencourt da Cruz et al., 2008*). To understand how such different mutations could result in similar outcomes, we first analyzed if *moody* loss would result in lysosomal material accumulation. Electron microscopy analyses demonstrated that unlike in *sws* mutant brains, no intracellular accumulations with extracellular material were observed upon *moody* loss (compare *Figure 3B and B′* and *Figure 5—figure supplement 2F*). Furthermore, no accumulation of endosomal-lysosomal pathway components such as Rab7 were detected within SPG cells of *moody* mutants (*Figure 5—figure supplement 2G and H*). At the same time, as previously described (*Babatz et al., 2018*; *Bainton et al., 2005*; *Schwabe et al., 2005*; *Li et al., 2021a*), we observed that in the absence of *moody*, SJs were formed, but they were disorganized (*Figure 5—figure supplement 2F*, arrow).

We compared in greater detail the SJ organization in both mutants using a molecular component of SJs, Neurexin IV (NrxIV). In comparison to the wild type, upon *sws* loss, SJs were not properly assembled and exhibited irregular membrane clusters and disruptions (*Figure 6A and C*). In contrast, the *moody* mutant exhibited a frayed SJ phenotype (*Figure 6B*). Since Moody coordinates the continuous organization of junctional strands in an F-actin-dependent manner, as a result of its loss, SJ

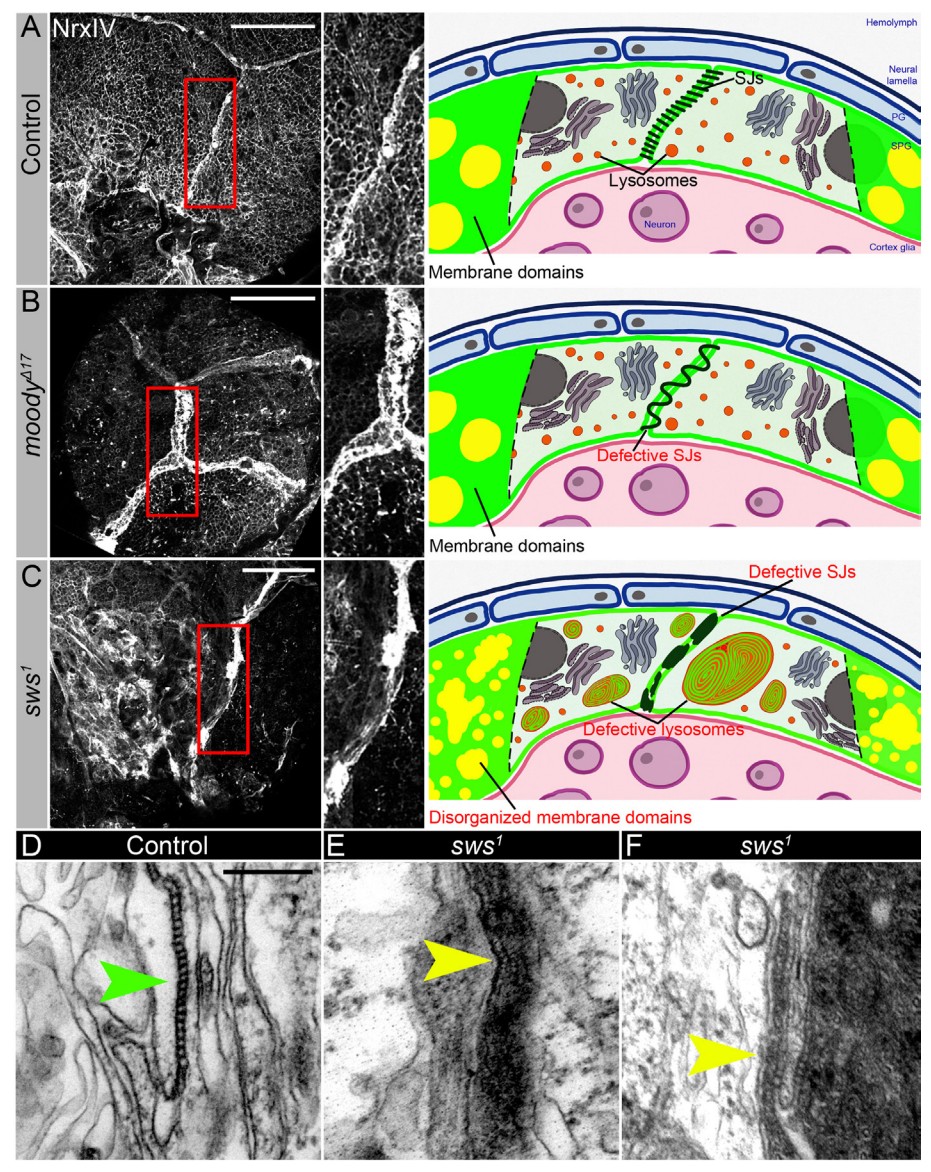

**Figure 6.** Septate junction and membrane domain organization in mutants with defective brain permeability barrier. (**A–C**) Adult brains stained with a septate junction marker Neurexin IV (NrxIV) (white). Scale bar: 50 μm. (**A**) In control (*Oregon R*) brain, septate junctions formed by subperineurial glia (SPG) glia are depicted as condensed and distinct strand. The scheme depicts the intact blood-brain barrier (BBB) formed by perineurial glia (PG) and SPG. The SPG cells establish well-formed septate junctions (SJs) and exhibit organized membrane domains. Furthermore, the lysosomes are fully functional. (**B**) In *moody*$^{\Delta C17}$ mutants, due to SPG membrane overgrowth, septate junctions are frayed. The scheme illustrates a defective BBB where the proper extension of septate junction strands during cell growth is impaired, resulting in increased permeability. However, despite this issue, the membrane domains remain well formed, and the lysosomes within the barrier continue to function effectively. (**C**) In *sws*$^1$ mutants, septate junctions and membrane domains are not properly organized. By analyzing SPG membranes in *sws* mutants, abnormal clustering of SJ proteins and disorganized membrane domains are observed. Furthermore, *sws*-deficient brains exhibit excessive storage of cellular material within lysosomes. The scheme shows that NTE/SWS-related lipid dysregulation is accompanied by dysfunctional lysosomes, impaired distribution of cell junction proteins, and disrupted organization of membrane domains in surface glia. (**D–F**) Electron microscopy images of the septate junction area at the surface of the control (*white*$^{1118}$, **D**) and *sws*$^1$ mutant (**E–F**) adult brains. Green arrowheads indicate septate junctions in control and yellow arrowheads indicate septate junctions in *sws*$^1$ mutant brains. Scale bar: 250 nm.

The online version of this article includes the following figure supplement(s) for figure 6:

**Figure supplement 1.** Lysosomal mutants show abnormal septate junction formation.

strands fail to extend properly during cell growth (*Figure 6C*). While the role of Moody in SJ formation is understood (*Babatz et al., 2018*; *Bainton et al., 2005*; *Schwabe et al., 2005*; *Li et al., 2021a*), the mechanism by which NTE/SWS may be involved in this process is unclear. The in-depth examination of cell junctional structures in *sws* mutants using electron microscopy revealed their improper assembly, characterized by the accumulation of irregular membrane clusters and disruptions in septa organization (*Figure 6E and F*, yellow arrowheads). Cell junctions are a special type of plasma membrane domain whose transmembrane proteins form a complex, mechanically stable multiprotein structure (*Giepmans and van Ijzendoorn, 2009*). The lipid component of cell junctions exhibits a typical membrane raft structure (*Lee et al., 2008*; *Shigetomi et al., 2023*; *Nusrat et al., 2000*; *Simons and Vaz, 2004*; *Mühlig-Versen et al., 2005*).

The main feature of membrane rafts is that they contain an enriched fraction of cholesterol and sphingolipids and are able to dynamically orchestrate specific membrane proteins involved in cell adhesion, signal transduction, protein transport, pathogen entry into the cell, etc. Since NTE/SWS regulates lipid membrane homeostasis, we hypothesized that it influences the composition of membrane rafts. Analysis of SPG membranes in *sws*-deficient brains shows abnormal clustering of SJs proteins and disorganized membrane domains, implying that NTE/SWS phospholipase plays a role in organizing SPG membrane architecture (*Figure 6C*). As lysosomes play a crucial role in lipid catabolism and transport, any disruptions in their function can have repercussions on cellular lipid homeostasis, thereby influencing the composition of membrane rafts. To investigate whether the observed SJ phenotype in *sws* mutants can be replicated by inducing lysosomal dysfunctions, we downregulated in SPG cells several key lysosomal genes: *moody>Dysb^{RNAi}*, *moody>Npc1a^{RNAi}*, *moody>Pldn^{RNAi}*, and *moody>spin^{RNAi}*. Significantly, the downregulation of any of these genes led to abnormal formation of SJs and membrane organization in SPG cells (*Figure 6—figure supplement 1A–E*). This suggests that the lysosomal control of membrane homeostasis has a significant impact on the appearance of SJs.

In summary, our data show that the phospholipase NTE/SWS plays a crucial role in lysosome biogenesis and organization of the architectural framework of BBB membranes. We propose that since NTE/SWS regulates lipid membrane homeostasis, its loss results in the disruption of membrane rafts, which includes SJs, leading to brain barrier permeability. As a result, the inflammatory response accompanied by the accumulation of FFAs is activated in mutant brains, leading to progressive neurodegeneration that can be alleviated by the use of anti-inflammatory drugs.

## Discussion

The physiological functions of the BBB, maintaining and protecting the homeostasis of the CNS, are evolutionarily conserved across species (*Bundgaard and Abbott, 2008*). Even though there is already plenty of evidence connecting BBB dysfunction to neurodegenerative diseases, the underlying mechanism is not fully understood. The BBB is formed by microvascular endothelial cells lining the cerebral capillaries penetrating the brain and spinal cord of most mammals and other organisms with a well-developed CNS (*Kadry et al., 2020*). Interestingly, NTE is highly expressed not only in the nervous system but also in endothelial cells, suggesting that BBB might be affected upon NTE-associated neurodegenerations (The Human Protein Atlas – https://www.proteinatlas.org/ENSG00000032444-PNPLA6/single+cell+type).

Here, we made an intriguing discovery regarding the presence of NTE/SWS in the surface glia responsible for forming the BBB, where it plays a crucial role in ensuring the selective permeability of the BBB and the proper organization of surface glia. Moreover, here we discovered that NTE/SWS-associated neurodegeneration is accompanied by abnormal membrane accumulation within defective lysosomes, indicating importance of NTE/SWS in proper function of lysosomes. It has been demonstrated for some LSDs, for example, Krabbe's disease, to be pathologically characterized by rapidly progressive demyelination of the CNS and PNS and accumulation of macrophages in the demyelinating lesions (*Kondo et al., 2005*). Considering that NTE/SWS is involved in the maturation of non-myelinating Schwann cells during development and de/remyelination after neuronal injury (*McFerrin et al., 2017*), it suggests that lysosomal function of NTE/SWS might be essential for proper myelination in vertebrates. Interestingly, we found that loss of *sws* or its downregulation in barrier-forming glia led to accumulations of Rab7 and CathepsinL in these cells, demonstrating that NTE/SWS-associated neuropathies might be additionally characterized by excessive storage of cellular material in lysosomes. Importantly, neuroinflammation has been reported in several LSDs. The most

abundant lysosomal proteases, Cathepsins have been shown to contribute to neuroinflammation as well as to induce neuronal apoptosis (*Tschopp and Schroder, 2010*).

Over the past few years, there has been a growing appreciation of the organizing principle in cell membranes, especially within the plasma membrane, where such domains are often referred to as 'lipid rafts'. Such lipid rafts were defined as transient, relatively ordered membrane domains, the formation of which is driven by lipid-lipid and lipid-protein interactions (*Sezgin et al., 2017*). Previously, it has been demonstrated that NTE/SWS is crucial for membrane lipid homeostasis, and *sws* mutants exhibit increased levels of phosphatidylcholine (*Mühlig-Versen et al., 2005*). Phosphatidylcholine, a key component of most organellar membranes, possesses an amphiphilic nature, enabling it to energetically self-assemble into continuous bilayers (*Yang et al., 2018*). This ability to spontaneously self-organize can explain the appearance of multilayered membrane structures in the lysosomes of *sws* mutants. Furthermore, phosphatidylcholine plays a vital role in generating spontaneous curvature, essential for membrane bending and tubulation in vesicular transport processes within the cell (*Epand and Epand, 1994*). Therefore, abnormal levels of phosphatidylcholine may impact the lysosome fission and fusion steps, leading to the accumulation of defective lysosomes in *sws* mutants. Since lysosomes are involved in lipid catabolism and transport, disruptions in their function can additionally affect cellular lipid homeostasis (*Thelen and Zoncu, 2017*). Consequently, alterations in lipid composition due to abnormal NTE/SWS phospholipase function and defective lysosomes in *sws* mutant cells could affect the constitution of the plasma membrane and its ability to form lipid-driven membrane rafts. Lipid rafts are characterized by the clustering of specific membrane lipids through spontaneous separation of glycolipids, sphingolipids, and cholesterol in a liquid-ordered phase (*Grassi et al., 2020*). Their assembly dynamics depend on the relative availability of different lipids and membrane proteins (*Simons and Vaz, 2004*). Lipid rafts play significant roles in multiple cellular processes, including signaling transduction (*Sezgin et al., 2017*). Interestingly, tight junctions are considered as raft-like membrane compartments (*Nusrat et al., 2000*), as they represent membrane microdomains crucial for the spatial organization of cell junctions and regulation of paracellular permeability (*Lee et al., 2008*; *Shigetomi et al., 2023*). Therefore, we propose that abnormal organization of tight junctions in the SPG cells of *sws* mutants is caused by abnormal organization of plasma membrane domains.

Lysosomes play an essential role in the breakdown and recycling of intracellular and extracellular material, including lipids, proteins, nucleic acids, and carbohydrates. Any dysfunction of lysosomal system components has catastrophic effects and leads to a variety of fatal diseases (*Udayar et al., 2022*). LSDs are often linked to changes in plasma membrane lipid content and lipid raft stoichiometry (*Domon et al., 2011*; *Vainio et al., 2005*), inflammation (*Seehafer et al., 2011*; *DiRosario et al., 2009*), and ER stress responses (*Kim et al., 2006*; *Tessitore et al., 2004*). In the past few years, treatments for LSDs were only able to deal with signs and symptoms of the disorders. One possible approach is to identify an available source for the deficient enzyme using therapeutic methods such as bone marrow transplantation, enzyme replacement therapy (ERT), substrate reduction therapy, chemical chaperone therapy, and gene therapy. At the present time, such strategies are aimed at relieving the severity of symptoms or delaying the disease's progression, yet do not provide a complete cure (*Sheth and Nair, 2020*). However, since we and others *Sujkowski et al., 2015* have shown that overexpression of human NTE can ameliorate mutant phenotype, it can be speculated that, depending on the causative mutation, ERT might be an option as treatment of NTE/SWS-related disorders.

It has been demonstrated that the ER establishes contacts between its tubules and late endosomes/lysosomes, visualized in unpolarized cells as well as in neurons derived from brain tissue. Moreover, disruption of ER tubules causes accumulation of enlarged and less-motile mature lysosomes in the soma, suggesting that ER shape and proper function orchestrate axonal late endosome/lysosome availability in neurons (*Özkan et al., 2021*; *Wu et al., 2017*). Considering the ER localization of NTE/SWS in the cell, we propose that abnormal lipid composition in the membrane upon *sws* loss has a significant effect on lysosome structure and functions. Furthermore, ER forms contact sites with plasma membrane through vesicle-associated membrane protein-associated protein VAP (*Li et al., 2021b*). Loss of VAP results in neurodegeneration, such as sporadic amyotrophic lateral sclerosis or Parkinson's disease (*Kun-Rodrigues et al., 2015*; *Anagnostou et al., 2010*). Mitochondria-ER contact sites play a crucial role in many vital cellular homoeostatic functions, including mitochondrial quality control, lipid metabolism, calcium homeostasis, unfolded protein response, and ER stress. Disruptions

in these functions are commonly observed in neurodegenerative disorders like Parkinson's disease, Alzheimer's disease, and amyotrophic lateral sclerosis (*Wilson and Metzakopian, 2021*). Interestingly, knockdown of *sws* in neurons reduces mitochondria number in the brain and in wing axons (*Melentev et al., 2021*). NTE/SWS-deficient animals show activation of ER stress response, characterized by elevated levels of GRP78 chaperone and increased splicing of XBP, an ER transcription factor that triggers transcriptional ER stress responses. Neuronal overexpressing XBP1 and treating flies with tauroursodeoxycholic acid (TUDCA), a chemical known to attenuate ER stress-mediated cell death, alleviated locomotor deficits and neurodegeneration in *sws* mutants assayed by vacuolization area (*Sunderhaus et al., 2019*). Reduced levels of sarco/endoplasmic reticulum $Ca^{2+}$ ATPase observed in *sws* mutants were linked to disrupted lipid compositions as well. Promoting cytoprotective ER stress pathways may provide therapeutic relief for NTE-related neurodegeneration and motor symptoms (*Sunderhaus et al., 2019*).

Moreover, we found that BBB disruption is accompanied by elevated levels of FFAs, involved in multiple extremely important biological processes. Fatty acids are locally produced in the endothelium and later are transported inside the brain across the BBB (*Pifferi et al., 2021*). We discovered that *Drosophila* mutants with leaky BBB showed upregulated levels of such fatty acids as palmitoleic, oleic, linoleic, linolenic, arachidonic, and eicosapentaenoic acids, suggesting abnormal metabolism of unsaturated fatty acids upon barrier dysfunction. In particular, *sws* loss results in increased levels of some saturated FFAs, including palmitic and stearic acids. FFAs or non-esterified fatty acids are known to be significant sources of ROS, which lead to the event of oxidative stress (*Soardo et al., 2011*), resulting in lipotoxicity associated with ER stress, calcium dysregulation, mitochondrial dysfunction, and cell death (*Ly et al., 2017*). Previously it has been demonstrated ROS accumulation and activated ER stress response upon *sws* loss in neurons and glia (*Melentev et al., 2021*; *Ryabova et al., 2021*; *Sunderhaus et al., 2019*), which might be a result of increased levels of FFAs. In addition, neuronal *sws* knockdown results in the upregulation of antioxidant defense genes (*Melentev et al., 2021*). We found that BBB breakdown is accompanied by abnormal fatty acids metabolism, and rapamycin can suppress the abnormal glial phenotype formed in BBB *Drosophila* mutants. Interestingly, saturated FFAs have been shown to lead to target of rapamycin (mTOR) complex 1 activation and cell apoptosis in podocytes (*Yasuda et al., 2014*). Moreover, rapamycin significantly diminishes FFA-induced podocyte apoptosis (*Yasuda et al., 2014*), supporting its potential ability to suppress possible outcomes of FFA upregulation in the *Drosophila* brain, thus improving glial phenotype in mutants with BBB breakdown.

PUFAs are known to be primary precursors of lipid mediators that are abundant immunomodulators (*Kwon et al., 2020*). Lipid mediators are signaling molecules, such as eicosanoids, and are implicated in inflammation. More recently, lipid molecules that are pro-inflammatory, and those involved in the resolution of inflammation have become important targets of therapeutic intervention in chronic inflammatory conditions. According to published research, PUFAs' metabolism was additionally associated with Alzheimer's disease and dementia (*van der Lee et al., 2018*; *Rao et al., 2017*). The focus of particular interest has recently been on the PUFAs' involvement in the continued inflammatory response because, in contrast to acute inflammation, chronic inflammatory processes within the CNS are crucial for the development of brain pathologies (*Regulska et al., 2021*; *Funk, 2001*). In our study, we found that brain permeability barrier breakdown is accompanied by abnormal fatty acids metabolism and that an aspirin analogue – an NSAID – showed the best ability to suppress abnormal glial phenotype, indicating that activated inflammatory response possibly plays an important role in maintaining a healthy brain barrier. Thus, feedback signaling loop exists between the condition of the brain permeability barrier, lipid metabolism, and the extent of inflammation. According to the World Health Organization (WHO), the current decade is considered the Decade of Healthy Aging. As the speed of population aging is accelerating worldwide, the proportion of older people will increase from one in eight people aged 60 years or over in 2017 to one in six by 2030 and one in five by 2050 (*Keating, 2022*; *Rudnicka et al., 2020*). Globally, there is a little evidence that older people today are in better health than previous generations (https://www.who.int/home/cms-decommissioning). If people who enter extended age of life are in good health, they will continue to participate and be an integral part of families and communities and will strengthen societies; however, if the added years are dominated by poor health, social isolation or dependency on care, the implications for older people and for society are much more negative. Therefore, aging of the world population has become one of the

most important demographic problems/challenges of modern society. Moreover, the global strategy on aging and health of the older population includes not only treating but also preventing some of the world's leading age-related diseases using biomarkers as indicators of any aspects of health change (*Crimmins et al., 2008*). Unfortunately, most neurodegenerative diseases in humans currently have no cure, and only palliative care is available. Current research is primarily focused on promoting the development of therapies that can prevent the onset of a number of age-related neurodegenerative diseases. Specific and effective treatments are urgently needed. However, their advance hinges upon a deeper understanding of the molecular mechanisms underlying progressive neurodegeneration. Understanding the molecular mechanisms of inflammaging activated by abnormal fatty acid metabolism and testing new and available drugs in a model organism such as *Drosophila* may help us to promote the use of anti-inflammatory therapy and dietary supplements for neurodegeneration and get closer to preventing and curing the diseases that lead to malfunctions in the aged brain.

## Materials and methods

### *Drosophila* stocks

Fly stocks were maintained at 25°C on a standard cornmeal-agar diet in a controlled environment (constant humidity and light-dark cycle). As controls *OregonR* and $w^{1118}$ lines were used. The $sws^1$ mutant and the *UAS-sws* lines were gifts from *Kretzschmar et al., 1997*. To obtain *sws* transheterozygotes, $sws^1$ and $sws^4$, obtained from Bloomington Drosophila Stock Center (BDSC 28121), mutant alleles were used. To express transgenes in an *sws*-dependent manner, an *sws* driver line (*sws-Gal4*), obtained from the Kyoto Stock Center (104592), was used. $y^* w^* P\{GawB\}sws^{NP4072}/FM7c$ line was created using the strategy of the Gal4 enhancer trap element P{GawB} insertion (*Brand and Perrimon, 1993*; *Hayashi et al., 2002*). To define an expression pattern of the driver lines, a *UAS-nlsLacZ, UAS-CD8::GFP* transgenic line, kindly donated by Frank Hirth, was used. To induce human NTE gene expression, a *UAS-hNTE* transgenic line (kindly donated by Robert Wessells) was used. To downregulate *sws* expression, $UAS$-$sws^{RNAi}$ (BDSC 61338) was used. Glia-specific Gal4 driver lines – *repo-Gal4, UAS-CD8::GFP/TM6B, Gliotactin-Gal4, UAS-CD8::GFP*, and *moody-Gal4, UAS-CD8::GFP* – were gifts from Mikael Simons. A neuronal Gal4 driver, *nSyb-Gal4* was obtained from BDSC (BDSC 51945). In addition, to phenocopy *sws* loss-of-function in the nervous system, a double driver line was generated (*repo-Gal4, nSyb-Gal4, UAS-CD8::GFP/TM6B, Sb),* which allowed the expression of the transgenes in both neuronal and glial cells. To induce *sws* downregulation in glia after the BBB was formed, we used *tub-Gal80^{ts};repo-Gal4/TM6B* driver line. The $moody^{AC17}$ mutant was a gift from Christian Klämbt. To downregulate *moody* expression, $UAS$-$moody^{RNAi}$ (BDSC 66326) was used. $UAS$-$Dysb^{RNAi}$ (BDSC67316), $UAS$-$Npc1a^{RNAi}$ (BDSC37504), $UAS$-$Pldn^{RNAi}$ (BDSC67884), $UAS$-$spin^{RNAi}$ (BDSC27702) lines were used to analyze SJs of lysosomal storage mutants. *srp(Hemo)3xmCherry* line (kindly donated by Angela Giangrande) was used to analyze the macrophage entry through the BBB.

### Histology of *Drosophila* brains

For analysis of adult brain morphology, 7 μm paraffin-embedded sections were cut from fly heads. To prepare *Drosophila* brain sections, the fly heads were immobilized in collars in the required orientation and fixed in Carnoy fixative solution (6:3:1=ethanol:chloroform:acetic acid) at 4°C overnight. Tissue dehydration and embedding in paraffin was performed as described previously (*Kucherenko et al., 2010*). Histological sections were prepared using a Hyrax M25 (Zeiss) microtome and stained with hematoxylin and eosin as described previously (*Shcherbata et al., 2007*). All chemicals for these procedures were obtained from Sigma-Aldrich.

### Immunohistochemistry

Fly brains of 1- and 15-day-old animals were dissected in 1× phosphate buffered saline (1× PBS) and then fixed in 4% formaldehyde diluted in 1× PBS for 20 min at room temperature. Next, brains were washed with PBT (0.2% Triton X-100 in 1× PBS) four times, followed by block with PBTB (2 g/L bovine serum albumin, 5% normal goat serum, 0.5 g/L sodium azide) for 1 hr at room temperature and then incubated at 4°C with primary antibodies diluted in PBTB on nutator overnight. The following day, samples were washed with 1× PBT four times followed by block for 1 and 2 hr incubation with secondary antibodies at room temperature. Next, samples were washed four times with

PBT (one of the washes contained DAPI to mark nuclei). Lastly, medium (70% glycerol, 3% n-propyl gallate in 1× PBS) was added to samples for later mounting on the slides. The following primary antibodies were used: mouse anti-Repo (1:50), mouse anti-CoraC (1:50), and mouse anti-Rab7 (1:50), rat anti-DE-Cadherin (1:50) from the Developmental Studies Hybridoma Bank (DSHB); chicken anti-GFP (#ab13970, 1:1000) and rabbit anti-mCherry (#ab167453, 1:1000) from Abcam; mouse Anti-β-Galactosidase (#Z3781, 1:200) from Promega; rabbit anti-SWS (1:1000 from Doris Kretzschmar); mouse anti-CathepsinL (#1515-CY-010, 1:400) from R&D Systems; rabbit anti-NrxIV (1:1000 from Christian Klämbt); mouse anti-NimC1 (1:300 from István Andó). The following secondary antibodies were used: goat anti-chicken Alexa 488 (1:500), goat anti-rat Alexa 488 (1:500), goat anti-rat Alexa 647 (1:500), goat anti-rabbit Alexa 488 (1:500), and goat anti-rabbit Alex 568 (1:500) from Thermo Fisher Scientific; goat anti-mouse IgG2a Cy3 (1:400), goat anti-mouse IgG1 647, and goat anti-mouse IgG1 Cy3 (1:500) from Jackson ImmunoResearch Laboratory. For visualization of cell nuclei, DAPI dye was used (1:1000, Sigma). Samples were analyzed using a confocal microscope (Zeiss LSM 700). For making figures, Adobe Photoshop software was used.

## RNA preparation and real-time qPCR

Total RNA was extracted from 15- and 30-day-old fly brains using Trizol reagent (Invitrogen) following the manufacturer's protocol. To detect *sws* mRNA, the forward and reverse primers AGACATACGCCG TGAATACCG and GCGACGACTGTGTGGACTTG, respectively, were used. To detect expression of innate immunity factors, the following forward and reverse primers were used: *Attacin A* (forward and reverse primers CACAACTGGCGGAACTTTGG and AAACATCCTTCACTCCGGGC, respectively), *Cecropin A* (forward and reverse primers AAGCTGGGTGGCTGAAGAAA and TGTTGAGC GATTCCCAGTCC, respectively), and *Diptericin* (forward and reverse primers TACCCACTCAATCTTC AGGGAG and TGGTCCACACCTTCTGGTGA, respectively). As an endogenous control for qPCRs, Ribosomal Protein L32 (RpL32) with the following forward and reverse primers AAGATGACCATC CGCCCAGC and GTCGATACCCTTGGGCTTGC, respectively, was used. The threshold cycle (CT) was defined as the fractional cycle number at which the fluorescence passes a fixed threshold. The $\Delta$CT value was determined by subtracting the average RpL32 mRNA CT value from the average tested CT value of target mRNA, correspondingly. The $\Delta\Delta$CT value was calculated by subtracting the $\Delta$CT of the control sample from the $\Delta$CT of the experimental sample. The relative amounts of miRNAs or target mRNA is then determined using the expression $2^{-\Delta\Delta CT}$.

## Permeability assay

Flies were injected into the abdomen with a solution containing 10 kDa dextran dye labeled with Texas Red (#D1864) from Molecular Probes. Flies were then allowed to recover for more than 12 hr before the dissection, followed by the analysis for dextran dye presence in the brain. Fly heads of 15-day-old animals were dissected in 1× PBS and then fixed in 4% formaldehyde diluted in 1× PBS for 1 hr at room temperature. Then fly brains were dissected in 1× PBS and fixed in 4% formaldehyde diluted in 1× PBS for 20 min at room temperature. Next, brains were washed with PBT (0.2% Triton X-100 in 1× PBS) four times, followed by block with PBTB (2 g/L bovine serum albumin, 5% normal goat serum, 0.5 g/L sodium azide) for 1 hr at room temperature and then washed two times with PBT (one of the washes contained DAPI to mark nuclei). Lastly, medium (70% glycerol, 3% n-propyl gallate in 1× PBS) was added to samples for later mounting on the slides.

## In vivo *Drosophila* treatments

TUDCA (#580549), 4-PBA (#567616), valsartan (#PHR1315), fenofibrate (#F6020), sodium salicylate (#S3007), rapamycin (#R0395), deferoxamine mesylate salt (#D9533), liproxstatin-1 (#SML1414), and sphingosine (#860025P) from Sigma-Aldrich were added to 5% glucose solution at a final concentration shown by *Supplementary file 5*. Then, these glucose-dissolved components were fed by micropipettes to the flies that were kept for 14 days on a diet food without any sugar. To visualize the uptake of chemicals, solutions were also colored by 2.5% wt/vol of Brilliant Blue (#80717) from Sigma-Aldrich.

## Extraction and derivatization of FFAs from flies

Accurately weighed heads of 15-day-old flies were treated with 1000 µL aliquots of acetonitrile in autosampler glass vials (1.8 mL), the samples were sealed and vortexed several times and then stored

in a refrigerator overnight (4°C). Next day, the samples were warmed up to room temperature and centrifuged (10 min, 3345×*g*, 4°C). Aliquots (950 µL) of the clear supernatants were decanted carefully transferred to autosampler glass vials (1.8 mL). The samples were spiked with 10 µL aliquots of a 1000 µM stock solution of sterculic acid (C19H34O2; 10 nmol; 8-cyclopropen-octadecenoic acid corresponds to C19:1) which served as the internal standard (IS) for all FFAs. The solvent was evaporated entirely under a stream of nitrogen gas. The solid residues were reconstituted in anhydrous acetonitrile (100 µL). Then, 10 µL Hünig base (*N*,*N*-diisopropylethylamine) and 10 µL 33 vol% pentafluorobenzyl (PFB) bromide in anhydrous acetonitrile were added. Subsequently, the FFAs were derivatized by heating for 60 min at 30°C to generate the PFB esters of the FFAs. Solvents and reagents were evaporated to dryness under a stream of nitrogen gas. The residues were treated with 1000 µL aliquots of toluene and the derivatives were extracted by vortex-mixing for 120 s. After centrifugation (10 min, 3345×*g*, 4°C), 300 µL aliquots of the clear and colorless supernatants were transferred into microvials placed in autosampler glass vials (1.8 mL) for GC-MS analysis. A standard control sample containing 1 mL acetonitrile, 1 µL 10 mM arachidonic acid (C20:4, 10 nmol), and 10 µL 1 mM IS (10 nmol) was derivatized as described above for the fly samples after were evaporated to dryness under a stream of nitrogen gas. After centrifugation (10 min, 3345×*g*, 4°C), 100 µL of the clear and colorless supernatant were transferred into an autosampler glass vial (1.8 mL), diluted with toluene (1:10, vol/vol), and subjected to GC-MS analysis as described below.

## GC-MS analysis of FFAs from flies

GC-MS analyses were performed on a GC-MS apparatus consisting of a single quadrupole mass spectrometer model ISQ, a Trace 1210 series gas chromatograph, and an AS1310 autosampler from Thermo Fisher (Dreieich, Germany). A fused-silica capillary column Optima 17 (15 m length, 0.25 mm ID, 0.25 µm film thickness) from Macherey-Nagel (Düren, Germany) was used. Aliquots of 1 µL were injected in the splitless mode. Injector temperature was kept at 280°C. Helium was used as the carrier gas at a constant flow rate of 1.0 mL/min. The oven temperature was held at 40°C for 0.5 min and ramped to 210°C at a rate of 15°C/min, and then to 320°C at a rate of 35°C/min. Interface and ion-source temperatures were set to 300°C and 250°C, respectively. Electron energy was 70 eV and electron current 50 µA. Methane (constant flow rate of 2.4 mL/min) was used as the reactant gas for negative-ion chemical ionization. The electron multiplier voltage was set to 1300 V. Authentic commercially available reference compounds were used to determine the retention times of the derivatives and to generate their mass spectra. The selected ions [M−PFB] − with mass-to-charge (*m/z*) ratios and retention times of the derivatives are summarized in *Supplementary file 6b*. Quantitative measurements were performed by selected-ion monitoring (SIM) of the ions listed in *Supplementary file 6b* with a dwell time of 50 ms and SIM width of 0.5 amu for each ion in three window ranges. The results of the GC-MS analyses of the control standard sample that contained 10 nmol arachidonic acid and 10 nmol IS are summarized in *Supplementary file 6c*. The highest peak area ratio of FFA to the internal standard (FFA/IS) was obtained for arachidonic acid (0.098). This in accordance with the ratio observed in the standard curve (*Figure 5—figure supplement 3A*). A lower FFA/IS was obtained for palmitic acid (0.026). As palmitic acid was not externally added to the control standard sample, it is assumed that palmitic fatty acid is ubiquitously present as a contamination in the laboratory materials. An FFA/IS value of 0.027 was obtained for a fatty acid, which co-elutes with nonadecanoic acid (C19:0). As this fatty acid was not externally added to the control standard sample nor it is expected to be a laboratory contamination, it can be hypothesized that it is a contamination in the commercially available preparation of the IS which is a quasi C19:0 fatty acid. The FFA/IS values of the other FFAs are remarkably lower (<0.0065), which suggest that they cannot be considered as appreciable contaminations (*Figure 5—figure supplement 3B*).

## Transmission electron microscopy

After dissection, brains of 15-day-old flies were fixed overnight immediately by immersion in 150 mM HEPES containing 1.5% glutaraldehyde and 1.5% formaldehyde at pH 7.35. Preparation for transmission electron microscopy was done as described (*Mariani et al., 2022*). Imaging was done in a Zeiss EM 900 at 80 kV, equipped with a side-mount CCD camera (TRS).

## Quantification and statistical analysis

To analyze the activation of the inflammatory response, real-time qPCR analysis of AMPs mRNA levels from heads of each genotype was performed. AVE ± SEM was calculated. The experiments were performed in at least two biological replicates for each genotype. Two-tailed Student's tests were used to test for statistical significance (*$p < 0.05$, **$p < 0.005$, ***$p < 0.001$, see *Supplementary file 1*).

To analyze the frequency of brain hemispheres with defective brain surfaces, Z-stack confocal images of the entire adult brain were captured. The brain surface was identified by CoraC expression. The numbers of brain hemispheres exhibiting a normal brain surface, those containing lesions, or those with both lesions and membrane clusters on the brain surface were quantified. For the comparison of observed phenotypes, two-way tables and chi-squared tests were used (*$p < 0.05$, **$p < 0.005$, ***$p < 0.001$, see *Supplementary file 2*).

To assess the frequency of brain hemispheres with the accumulation of Rab7-positive or CathepsinL-positive structures in surface glia, Z-stack confocal images of the entire adult brain were captured. The surface glia were identified by *moody-Gal4, UAS-CD8::GFP* expression. The numbers of brain hemispheres with Rab7 or CathepsinL accumulation in the surface glia were quantified. For the comparison of observed phenotypes, two-way tables and chi-squared tests were used (*$p < 0.05$, **$p < 0.005$, ***$p < 0.001$, see *Supplementary file 3*).

To analyze the frequency of brain hemispheres with a permeable BBB, Z-stack confocal images of the entire adult brain were captured. The permeable BBB was identified by 10 kDa dextran dye labeled with Texas Red localization inside the fly brain. The numbers of brain hemispheres with a permeable BBB were quantified. All experiments were performed in at least three biological replicates for each genotype. For the comparison of observed phenotypes, two-way tables and chi-squared tests were used (*$p < 0.05$, **$p < 0.005$, ***$p < 0.001$, see *Supplementary file 4*).

To analyze the frequency of brain hemispheres with defective brain surfaces in in vivo *Drosophila* treatment assays, the brain surface was identified by CoraC expression. The numbers of brain hemispheres with formed lesions and membrane clusters on the brain surface were quantified. The reduction in the percentage of the glial phenotype, assayed by CoraC expression pattern in *sws* and *moody* mutants treated with different chemicals compared to untreated mutants, was quantified. All experiments were performed in at least three biological replicates for each genotype. For the comparison of observed phenotypes, two-way tables and chi-squared tests were used (*$p < 0.05$, **$p < 0.005$, ***$p < 0.001$, see *Supplementary file 5*).

To analyze the changes in FFA levels in fly mutant heads, GS-MS measurements of FFAs were performed. One-way ANOVA tests were used for statistical analysis (*$p < 0.05$, **$p < 0.005$, ***$p < 0.001$, see *Supplementary file 6*).

## Acknowledgements

We would like to thank Doris Kretzschmar, Mikael Simons, Christian Klämbt, Hugo Bellen, Angela Giangrande, István Andó, and Stanislava Chtarbanova-Rudloff for sharing flies and reagents with us. Marko Shcherbatyy for drawing a scheme. Christian Klämbt and Volkan Seyrantepe for contributions to phenotype description. All Shcherbata lab members for critical reading of the manuscript and helpful suggestions. This research was supported by the VolkswagenStiftung (grants 90218 and 97750), the German Research Foundation (DFG) grant numbers 521749003 and INST 192/574-1 FUGG, the Institutional Development Award (IDeA) from the National Institute of General Medical Sciences (NIGMS) of the National Institutes of Health (NIH) under grant numbers P20GM103423 and P20GM104318 (to the Mount Desert Island Biological Laboratory), and EMBO YIP.

## Additional information

### Funding

| Funder | Grant reference number | Author |
| --- | --- | --- |
| Volkswagen Foundation | 90218 | Halyna R Shcherbata |
| Volkswagen Foundation | 97750 | Halyna R Shcherbata |

| Funder | Grant reference number | Author |
|---|---|---|
| Deutsche Forschungsgemeinschaft | 521749003 | Halyna R Shcherbata |
| Deutsche Forschungsgemeinschaft | INST 192/574-1 FUGG | Halyna R Shcherbata |
| European Molecular Biology Organization | Young Investigator Programme | Halyna R Shcherbata |

The funders had no role in study design, data collection and interpretation, or the decision to submit the work for publication.

## Author contributions

Mariana I Tsap, Formal analysis, Validation, Investigation, Visualization, Methodology, Writing - original draft, Writing - review and editing; Andriy S Yatsenko, Formal analysis, Supervision, Validation, Investigation, Visualization; Jan Hegermann, Formal analysis, Investigation, Methodology; Bibiana Beckmann, Dimitrios Tsikas, Formal analysis, Investigation; Halyna R Shcherbata, Conceptualization, Data curation, Formal analysis, Supervision, Funding acquisition, Validation, Investigation, Visualization, Methodology, Writing - original draft, Project administration, Writing - review and editing

## Author ORCIDs

Mariana I Tsap http://orcid.org/0009-0006-0891-8504
Dimitrios Tsikas http://orcid.org/0000-0001-6320-0956
Halyna R Shcherbata http://orcid.org/0000-0002-3855-0345

## Decision letter and Author response

Decision letter https://doi.org/10.7554/eLife.98020.sa1
Author response https://doi.org/10.7554/eLife.98020.sa2

---

# Additional files

## Supplementary files

• Supplementary file 1. Relative mRNA levels. [a] – the ΔCT value is determined by subtracting the average CT value of endogenous control gene(Rpl32) from the average mRNA CT value. [b] –the calculation of ΔΔCT involves subtraction by the ΔCT calibrator value (ΔCT value in control). [c] – the range is given for relative levels determined by evaluating the expression: $2^{-\Delta\Delta CT}$. AVE ± SEM values are reported from experiments done in at least duplicates. Two-tailed Student's test was used to test for statistical significance. $p^a$ – compared to the relevant control. $p^b$ – compared to 15-day-old animals of the same genotype. $p^c$ – compared to $sws^1$ mutant of the same age.

• Supplementary file 2. NTE/SWS expression in the surface glia is important for the integrity of *Drosophila* blood-brain barrier (BBB). [a] – compared to control (*OR x w^1118*). [b] – compared to *Gal4-driver x OR*. [c] – compared to *Gal4-driver x UAS-sws^RNAi*. The values are reported from experiments done in triplicates. For statistical analyses of the observed phenotypes, two-way tables and chi-squared test were used.

• Supplementary file 3. NTE/SWS deficit in the surface glia results in the accumulation of Rab7- and CathepsinL-positive structures. [a] – compared to *Gal4-driver x OR* animals of the same age. [b] – compared to 1-day-old animals of the same genotype. The values are reported from experiments done in triplicates. For statistical analyses of the observed phenotypes, two-way tables and chi-squared test were used.

• Supplementary file 4. NTE/SWS deficit in the surface glia results in permeable blood-brain barrier (BBB). [a] – compared to control (*OR x w^1118*). [b] – compared to *Gal4-driver x OR*. [c] –compared to *Gal4-driver x UAS-sws^RNAi*. The values are reported from experiments done in triplicates. For statistical analyses of the observed phenotypes, two-way tables and chi-squared test were used.

• Supplementary file 5. The effect of treatment with different anti-inflammatory substances and stress suppressors on the frequency of the surface glia phenotype in *sws* and *moody* mutants. [a] – compared to *sws^1* (no drug treatment)[b] – compared to *moody^ΔC17* (no drug treatment). The values are reported from experiments done in triplicates. For statistical analyses of the observed phenotypes, two-way tables and chi-squared test were used.

• Supplementary file 6. Mutants with defective BBB show upregulated levels of free fatty acids (FFA). (a) *sws* and *moody* mutants show upregulated levels of free fatty acids (FFAs). For statistical analyses one-way ANOVA test was used. C14:1–9-cis-Tetradecenoic acid. C16:0 – Palmitic acid. C16:1 – Palmitoleic acid. C18:0 – Stearic acid. C18:1 – Oleic acid. C18:2 – Linoleic acid. C18:3 – α- and γ-Linolenic acid. C20:0 – Eicosanoic acid. C20:4 – Arachidonic acid. C20:5 – Eicosapentaenoic acid. (b) *sws* and *moody* mutants show upregulated levels of free fatty acids (FFAs). Summary of the ions monitored in the selected-ion monitoring (SIM) modeSIM#1 (12.00–14.50 min): *m/z* 197.4, 199.4, 225.4, 227.4, 253.4, 255.4, 267.4, 269.4. SIM#2 (14.50–15.00 min): *m/z* 281.4, 283.4, 279.4, 295.4, 297.4. SIM#3 (15.00–17.00 min): *m/z* 301.4, 303.4, 309.4, 311.4, 325.4, 337.4, 339.4, 365.4, 367.4.

• MDAR checklist

## Data availability

All data generated or analysed during this study are included in the manuscript and supporting files.

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

# Appendix 1

## Appendix 1—key resources table

| Reagent type (species) or resource | Designation | Source or reference | Identifiers | Additional information |
|---|---|---|---|---|
| Antibody | Anti-Repo (mouse monoclonal) | Developmental Studies Hybridoma Bank | #8D12 | IF(1:50) |
| Antibody | Anti-CoraC (mouse monoclonal) | Developmental Studies Hybridoma Bank | #C566.9 | IF(1:50) |
| Antibody | Anti-Rab7 (mouse monoclonal) | Developmental Studies Hybridoma Bank | #AB2722471 | IF(1:50) |
| Antibody | Anti-DE-Cad (rat monoclonal) | Developmental Studies Hybridoma Bank | #DCAD2 | IF(1:50) |
| Antibody | Anti-GFP (chicken polyclonal) | Abcam | #ab13970 | IF(1:1000) |
| Antibody | Anti-mCherry (rabbit polyclonal) | Abcam | #ab167453 | IF(1:1000) |
| Antibody | Anti-β-Galactosidase (mouse monoclonal) | Promega | #Z3781 | IF(1:200) |
| Antibody | Anti-CathepsinL (mouse) | R&D Systems | #1515-CY-010 | IF(1:400) |
| Antibody | Anti-NrxIV (rabbit polyclonal) | Gift from Christian Klämbt | Anti-NrxIV | IF(1:1000) |
| Antibody | Anti-SWS (rabbit polyclonal) | Gift from Doris Kretzschmar | Anti-SWS | IF(1:1000) |
| Antibody | Anti-NimC1 (mouse) | Gift from István Andó | Anti-NimC1 | IF(1:300) |
| Antibody | Anti-chicken Alexa 488 (goat polyclonal) | Thermo Fisher Scientific | #A-11039 | Secondary antibody IF(1:500) |
| Antibody | Anti-rat Alexa 488 (goat polyclonal) | Thermo Fisher Scientific | #A-11077 | Secondary antibody IF(1:500) |
| Antibody | Anti-rat Alexa 647 (goat polyclonal) | Thermo Fisher Scientific | #A-21247 | Secondary antibody IF(1:500) |
| Antibody | Anti-rabbit Alexa 488 (goat polyclonal) | Thermo Fisher Scientific | #A-11034 | Secondary antibody IF(1:500) |
| Antibody | Anti-rabbit Alexa 568 (goat polyclonal) | Thermo Fisher Scientific | #A-11011 | Secondary antibody IF(1:500) |
| Antibody | Anti-mouse IgG2a Cy3 (goat polyclonal) | Jackson ImmunoResearch | #115-165-206 | Secondary antibody IF(1:400) |
| Antibody | Anti-mouse IG1 Cy3 (goat polyclonal) | Jackson ImmunoResearch | #115-165-205 | Secondary antibody IF(1:500) |
| Antibody | Anti-mouse IgG1 647 (goat polyclonal) | Jackson ImmunoResearch | #115-605-205 | Secondary antibody IF(1:500) |
| Genetic reagent (*D. melanogaster*) | w[1118] | Bloomington *Drosophila* Stock Center | BDSC 5905 | Wild type strain |
| Genetic reagent (*D. melanogaster*) | Oregon-R | Bloomington *Drosophila* Stock Center | BDSC 5 | Wild type strain |
| Genetic reagent (*D. melanogaster*) | sws[1] | Gift from Doris Kretzschmar | sws[1] | sws[1]/FM7a(null mutant) |
| Genetic reagent (*D. melanogaster*) | sws[4] | Bloomington *Drosophila* Stock Center | BDSC 28121 | sws[4]/C(1)DX, y[1] w[1] f[1] (Amino acid replacement: G956D) |
| Genetic reagent (*D. melanogaster*) | UAS-sws | Gift from Doris Kretzschmar | UAS-sws | UAS-sws (sws gene under control of UAS promotor) |
| Genetic reagent (*D. melanogaster*) | sws-Gal4 | Kyoto Stock Center | 104592 | y* w* P{GawB}swsNP4072/FM7c |
| Genetic reagent (*D. melanogaster*) | UAS-nlsLacZ, UAS-CD8::GFP | Gift from Frank Hirth | UAS-nLacZ, UAS-GFP | UAS-nlsLacZ, UAS-CD8::GFP(nLacZ and GFP constructs under control of UAS promotor) |

*Appendix 1 Continued on next page*

*Appendix 1 Continued*

| Reagent type (species) or resource | Designation | Source or reference | Identifiers | Additional information |
|---|---|---|---|---|
| Genetic reagent (*D. melanogaster*) | UAS-hNTE | Gift from Robert Wessells | UAS-hNTE | w[1118]; p[PUAST]-hNTE/CyO(Human NTE undercontrol of UAS promotor) |
| Genetic reagent (*D. melanogaster*) | UAS-sws<sup>RNAi</sup> | Bloomington *Drosophila* Stock Center | BDSC 61338 | y[1] v[1]; P{y[+t7.7] v[+t1.8]=TRiP.HMJ23229}attP40 (sws RNAi construct under control of UAS promotor) |
| Genetic reagent (*D. melanogaster*) | repo-Gal4, UAS-CD8::GFP/TM6B | Gift from Mikael Simons | repo-Gal4 | repo-Gal4, UAS-CD8::GFP/TM6B |
| Genetic reagent (*D. melanogaster*) | Gliotactin-Gal4, UAS-CD8::GFP | Gift from Mikael Simons | Gli-Gal4 | Gliotactin-Gal4, UAS-CD8::GFP |
| Genetic reagent (*D. melanogaster*) | moody-Gal4, UAS-CD8::GFP | Gift from Mikael Simons | moody-Gal4 | moody-Gal4, UAS-CD8::GFP |
| Genetic reagent (*D. melanogaster*) | nSyb-Gal4 | Bloomington *Drosophila* Stock Center | BDSC 51945 | y[1] w[1118]; P{y[+t7.7] w[+mC]=nSyb-GAL4.DBD::QF.AD}attP2 |
| Genetic reagent (*D. melanogaster*) | repo-Gal4, nSyb-Gal4, UAS-CD8::GFP/TM6B,Sb | This study | repo-Gal4, nSyb-Gal4 | repo-Gal4, nSyb-Gal4, UAS-CD8::GFP/TM6B,Sb |
| Genetic reagent (*D. melanogaster*) | tub-Gal80<sup>ts</sup>; repo-Gal4/TM6B | This study | tub-Gal80<sup>ts</sup>; repo-Gal4/TM6B | tub-Gal80<sup>ts</sup>; repo-Gal4/TM6B(temperature sensitive) |
| Genetic reagent (*D. melanogaster*) | moody<sup>ΔC17</sup> | Gift from Christian Klämbt | moody<sup>ΔC17</sup> | Null mutant |
| Genetic reagent (*D. melanogaster*) | UAS-moody<sup>RNAi</sup> | Bloomington *Drosophila* Stock Center | BDSC 66326 | y[1] sc[*] v[1] sev[21]; P{y[+t7.7] v[+t1.8]=TRiP.HMC06237}attP2(moody RNAi construct under control of UAS promotor) |
| Genetic reagent (*D. melanogaster*) | UAS-Dysb<sup>RNAi</sup> | Bloomington *Drosophila* Stock Center | BDSC 67316 | y[1] sc[*] v[1] sev[21]; P{y[+t7.7] v[+t1.8]=TRiP.HMC06420}attP40/CyO(Dysb RNAi construct under control of UAS promotor) |
| Genetic reagent (*D. melanogaster*) | UAS-Npc1a<sup>RNAi</sup> | Bloomington *Drosophila* Stock Center | BDSC 37504 | y[1] sc[*] v[1] sev[21]; P{y[+t7.7] v[+t1.8]=TRiP.HMS01646}attP40 (Npc1a RNAi construct under control of UAS promotor) |
| Genetic reagent (*D. melanogaster*) | UAS-Pldn<sup>RNAi</sup> | Bloomington *Drosophila* Stock Center | BDSC 67884 | y[1] sc[*] v[1] sev[21]; P{y[+t7.7] v[+t1.8]=TRiP.HMS05728}attP40 (Pldn RNAi construct under control of UAS promotor) |
| Genetic reagent (*D. melanogaster*) | UAS-spin<sup>RNAi</sup> | Bloomington *Drosophila* Stock Center | BDSC 27702 | y[1] v[1]; P{y[+t7.7] v[+t1.8]=TRiP.JF02782}attP2 (spin RNAi construct under control of UAS promotor) |
| Genetic reagent (*D. melanogaster*) | srp(Hemo) 3xmCherry | Gift from Angela Giangrande | srp(Hemo) 3xmCherry | srp(Hemo) 3xmCherry |
| Software, algorithm | Microsoft Excel | Microsoft | Microsoft Excel | |
| Software, algorithm | Adobe Photoshop | Adobe | Adobe CC | |
| Software, algorithm | Zen 2011 (black edition) | Carl Zeiss; **Emmenlauer et al., 2009** | Zen 2011 | |
| Software, algorithm | AlphaFold2 v1.5.2 | **Mirdita et al., 2022**; https://colab.research.google.com/github/sokrypton/ColabFold/blob/main/AlphaFold2.ipynb | AlphaFold2 | |
| Software, algorithm | The PyMOL Molecular Graphics System, v2.5.5 | Schrödinger, LLC; https://pymol.org/ | PyMol | |

*Appendix 1 Continued on next page*

*Appendix 1 Continued*

| Reagent type (species) or resource | Designation | Source or reference | Identifiers | Additional information |
|---|---|---|---|---|
| Software, algorithm | StepOne Software v2.3 | Applied Biosystems | StepOne | |
| Chemical compound, drug | TRIzol reagent | Invitrogen | #15596018 | |
| Commercial assay or kit | High Capacity cDNA Reverse Transcription kit | Applied Biosystems | #4368813 | |
| Commercial assay or kit | FastSYBR Green master mix | Applied Biosystems | #435612 | |
| Chemical compound, drug | TUDCA | Sigma-Aldrich | #580549 | Tauroursodeoxycholic acid |
| Chemical compound, drug | 4-PBA | Sigma-Aldrich | #567616 | 4-Phenylbutyric acid |
| Chemical compound, drug | Valsartan | Sigma-Aldrich | #PHR1315 | |
| Chemical compound, drug | Fenofibrate | Sigma-Aldrich | #F6020 | |
| Chemical compound, drug | Sodium salicylate | Sigma-Aldrich | #S3007 | |
| Chemical compound, drug | Rapamycin | Sigma-Aldrich | #R0395 | |
| Chemical compound, drug | Deferoxamine mesylate salt | Sigma-Aldrich | #D9533 | |
| Chemical compound, drug | Liproxstatin-1 | Sigma-Aldrich | #SML1414 | |
| Chemical compound, drug | Sphingosine | Sigma-Aldrich | #860025P | |
| Chemical compound, drug | Brilliant Blue | Sigma-Aldrich | #80717 | |
| Chemical compound, drug | Acetic acid | Sigma-Aldrich | #27225-1L-M | |
| Chemical compound, drug | Chloroform | Sigma-Aldrich | #288306-2L | |
| Chemical compound, drug | Glycerol | Sigma-Aldrich | #G6279-1L | |
| Chemical compound, drug | Sodium azide | Sigma-Aldrich | #S2002-25G | |
| Chemical compound, drug | Formaldehyde, 16% | Polysciences Inc | #18814-20 | Methanol free, ultra pure |
| Other | 10 kDa Dextran | Molecular Probes | #D1864 | Dye labeled with Texas Red |
| Other | DAPI stain | Sigma-Aldrich | #D9542-10MG | IF concentration used: 1 µg/mL |
| Other | Normal Goat Serum | Abcam | #ab7481 | |
| Other | Paraplast Plus | Sigma-Aldrich | #76258-1KG | |
| Other | Casein Blocking Buffer 10× | Sigma-Aldrich | #B6429-500ML | |

*Appendix 1 Continued on next page*

*Appendix 1 Continued*

| Reagent type (species) or resource | Designation | Source or reference | Identifiers | Additional information |
|---|---|---|---|---|
| Other | Hematoxylin Solution, Mayer's | Sigma-Aldrich | #MHS16-500ML | |
| Other | Eosin Y solution, aqueous | Sigma-Aldrich | #HT110232 | |
| Other | DPX Mountant for histology | Sigma-Aldrich | #06522-100ML | |
| Other | PBS buffer (10× Dulbecco's) | AppliChem | #A0965,9010 | |
| Other | LSM700 confocal laser-scanning microscope | Carl Zeiss | LSM700 | |
| Other | Hyrax M25 microtome | Carl Zeiss | Hyrax M25 | |
| Other | Zeiss EM 900 microscope | Carl Zeiss | Zeiss EM 900 | |
| Other | Step One Plus 96 well system | Applied Biosystems | Step One Plus | |
| Chemical compound, drug | Acetonitrile | Honeywell | #34851 | |
| Chemical compound, drug | Toluene | Supelco | #1.08325.1000 | |
| Chemical compound, drug | Pentafluoro-benzyl bromide | Sigma-Aldrich | #101052 | |
| Chemical compound, drug | Diisopropyl-ethylamine | Sigma-Aldrich | #496219 | |
| Commercial assay or kit | GC-MS ISQ | Thermo Fisher | Trace 1210 series | |
| Commercial assay or kit | Optima 17 | Macherey-Nagel | #MN726022.15 | |
| Chemical compound, drug | C12:0; C12:1; C14:0; C14:1; C16:0; C16:1; C17:0; C17:1; C18:0; C18:1; C18:2; C18:3; C20:4; C20:5; C21:0; C22:0; C24:0 | Merck (Darmstadt, Germany) | | |
| Chemical compound, drug | C19:1; C20:0; C21:1 | Larodan AB (Solna, Sweden) | | |
| Chemical compound, drug | Paraformaldehyde | Merck | #1.04005.1000 | |
| Chemical compound, drug | Glutaraldehyde | Merck | #1.04239.0250 | |
| Chemical compound, drug | Osmiumtetroxide | Electron Microscopy Sciences | #22400-56 | |
| Chemical compound, drug | HEPES | Roth | #7020.2 | |
| Chemical compound, drug | Acetone puriss.p.a. ACS reagent, reag.ISO 99, 5% | Sigma-Aldrich | #:32201-2.5 | |
| Chemical compound, drug | Agar 100 Premix Kit – Hard | Agar Scientific | #R1140 | |
| Chemical compound, drug | Tri-Natriumcitrat-Dihydrat | Merck | #1-06448.0500 | |
| Chemical compound, drug | Lead (II) nitrate for analysis | Merck | #1.07398.0100 | |

