## [Editor Report]

The study underscores the essential role of Neuropathy Target Esterase (NTE)/Swiss Cheese (SWS) in preserving the blood-brain barrier (BBB) integrity and links its dysfunction to symptoms akin to lysosomal storage diseases, elevated fatty acid levels leading to abnormal cellular architecture, and inflammation. It further elaborates on how a compromised BBB facilitates an inflammatory response and fatty acid accumulation, exacerbating neurodegenerative conditions. These important findings backed by solid evidence suggest that targeting inflammation and fatty acid dysregulation may offer therapeutic strategies for age-related neurodegeneration.

---

## [Decision Letter]

[Editors' note: this paper was reviewed by Review Commons.]

---

## [Author Response]

We would like to sincerely thank the reviewers for the positive evaluation of our work, careful reading of our manuscript, and helpful suggestions. In the revised version of our manuscript, we have introduced the proposed changes and added the new data based on the suggested experiments to address the reviewers’ concerns. We hope that this modified version of the manuscript is now acceptable for publication.

Reviewer #1 (Evidence, reproducibility and clarity (Required)):SummaryElucidating the cellular and molecular mechanisms underlying age-related neurodegeneration remains a key challenge for neurobiologists. In this manuscript, Mariana Tsap and colleagues in the team of Halyna Shcherbata focus on the function of the neuropathy target esterase NTE/Swiss Cheese (Sws) in the *Drosophila* brain. The authors use an elegant combination of genetics, light and electron microscopy, RT-qPCR and GS-MS mass spectrometry to determine the complex role of Sws in cellular blood brain barrier (BBB) integrity, the brain inflammatory response and fatty acid metabolism. The study provides a detailed characterisation as to how the loss of sws affects glial cell morphology in the BBB revealing abnormal membrane accumulations and tight junctions, and in consequence causing permeability issues. Importantly, they observed the upregulation of antimicrobial peptides in the brain, indicative of neuroinflammation, as well as of fatty acids, equally connected with the inflammatory response.Major commentsThe study provides a detailed and comprehensive characterization of the sws mutant phenotype, and in particular the role of this gene in blood-brain barrier forming glia.• The study connects neurodegeneration and inflammation, but also makes a particular point about "inflammaging". However, the age contribution has not been studied in detail. Indeed, the flies analyzed are 15 days old (according to the Material and Methods section, with the exception of Figure 1 where flies are 30 days old), and hence have not been compared with younger or older flies to make a point of age as evoked in the abstract, introduction or discussion. The authors should either add experiments comparing differently aged flies or de-emphasize this point to a brief consideration in the discussion. Instead, it would be very helpful to provide concise information about the current knowledge concerning the inflammatory response in the *Drosophila* brain.

We thank the reviewer for raising this point. The decision to use 15-day-old flies was made due to the high mortality of *sws* mutants after two weeks and because age-dependent character of *sws* neurodegeneration has been previously well described. As the reviewer suggested, now we also included old animals in our experiments to show a connection between age-dependent neurodegeneration and inflammation. We measured and compared the mRNA levels of expression of the antimicrobial peptides (AMPs) Attacin A, Cecropin A, and Diptericin in the heads of 15- and 30-day-old *sws* loss-of-function mutants, in the heads of flies that had *sws* downregulation only in SPG cells (*moody>sws^RNAi^*) and in the heads of flies expressing NTE/SWS in SPG cells in *sws* mutant background. We found that the expression levels of the antimicrobial peptides are increased in the age-dependent manner in the tested mutants. In addition, we found that the expression of NTE/SWS in SPG in *sws* mutant background reduces inflammatory response in aging animals (see Figure 5D). Also, as the reviewer suggested, we provide brief information on the current understanding of the inflammatory response in the *Drosophila* brain in the Introduction and Results sections.

• Related to this point, the authors convincingly show that sws is required in surface glia using rescue experiments. Nevertheless, all experiments rely on drivers and mutants that could cause the emergence of phenotypes during development. Thus, to strengthen the causative link between the breakdown of the BBB and the neuroinflammatory response, it would be helpful to consider an acute knock-down in adults after BBB formation has been completed.

To strengthen the causative link between the breakdown of the BBB and the neuroinflammatory response during adulthood, we performed qPCR analysis and measured the mRNA levels of the antimicrobial peptides Attacin A, Cecropin A, and Diptericin in the heads of flies with *sws* downregulation in glia cells induced after the blood-brain barrier was formed using the *Gal80^ts^* tool. We found that *sws* downregulation in glial cells during adulthood, after the BBB is formed, leads to the increased inflammatory response (Figure 5 —figure supplement 1E).

• To test the brain permeability barrier, the study uses a 10 KDa dextran permeability assay. Almost 25% of brain in controls show a leaky barrier. It would be helpful to describe the causes for this relatively high occurrence.

The observed relatively high occurrence of a leaky barrier phenotype in our control group may be attributed to our experimental procedure. We injected flies peritoneally and waited for over 12 hours before dissecting their brains for the permeability assay. Typically, such analyses are conducted after shorter periods, often around 2 hours. Additionally, we used Dextran with the smallest molecular weight (10kDa). The blood-brain barrier (BBB) is not 100% impermeable, and small molecules can gradually enter the brain over time. Recent studies have shown that this entry could be facilitated by endocytosis (Artiushin *et al.,* 2018), which could partially explain the presence of Dextran 10kDa in control brains. Considering this, using a larger Dextran (70kDa) in our experiments could have been more accurate. Importantly, we always compared mutants and controls that underwent identical treatment, dissection, and analysis. We conducted experiments in multiple biological replicates to accurately assess the significance of the differences between mutants and controls. Therefore, we are confident that the differences we observed between controls and mutant flies in the BBB permeability are significant. We included all relevant numbers and statistics for these experiments in Supplementary file 4.

• An important point in the study concerns the increase of free fatty acids as cause of the inflammatory response. The measurements were based on measurements of whole heads, which could include the hemolymph and fat body within the head in addition to brain. However, the causative relationship remains unclear and the question why a leaky blood brain barrier would increase the free fatty acid levels in the body or brain remains mainly an observation at the descriptive level. Here, it would be helpful to design an experiment, which could test the causative links or to modify the interpretation in scheme 6D and adjust the wording in the text.

We agree that the causative relationship between a leaky blood-brain barrier and increased free fatty acid levels in the body or brain is currently an observation at the descriptive level and that it would be important to investigate the correlation between a leaky blood-brain barrier, inflammation, and increased free fatty acid levels in greater detail in future studies. In the modified manuscript, we have changed the scheme in Figure 5G and adjusted the wording in the text.

• Related to this, how do the levels of AMP caused by a leaky BBB would compare to an elicited neuroinflammation by the presence of bacteria? The neuroinflammatory response can be accompanied by macrophage entry into the brain following AMP induction. Could the authors detect this response (which could be envisioned as manipulations include pupal development, provided macrophages would persist into adulthood)? This would make a strong point regardless of the outcome.

We thank the reviewer for suggesting this excellent experiment. To detect macrophage entry into the mutant brains, we used antibodies (NimC1) and *srp(Hemo)>mCherry* that label the macrophage cells. We found macrophages in the larval and adult *sws* mutant brains and also in adult brains upon downregulation of *sws* in SPG cells (Figure 5E-F and Figure 5 —figure supplement 1F-G). These data additionally support our hypothesis that a leaky BBB in *sws* mutants induces neuroinflammation, which is accompanied by macrophage entry into the brain following AMP expression.

• Expression of sws is determined using sws-Gal4 driving membrane-tethered GFP. As sws is expressed very widely and classical Gal4 lines tend to be active in the BBB, it is important to provide the exact information about the nature of this driver.

We appreciate the reviewer for bringing this to our attention. We have now included information about the line we used to express transgenes in a *sws*-dependent manner. Specifically, we utilized the *y*w*P{GawB}swsNP4072/FM7c* line (Kyoto Stock Center 104592), which was generated using the Gal4 enhancer trap element P{GawB} insertion strategy.

• The Material and Methods section should contain a proper Quantification and Statistical analysis section. In the Figures, it would be helpful to refer to the Table reporting sample numbers.

As the reviewer suggested, we have now included a Quantification and Statistical analysis section in the Materials and methods. Additionally, we ensured that all figure legends include a reference to the corresponding tables reporting sample numbers and statistics.

• In Figure 5, it would be important to indicate sample numbers, the nature of the error bar, and show data points together with columns.

We agree with the reviewer that it is important to report all sample numbers and statistics. We generated a new Supplementary file 1 for all qRT-PCT data, and Supplementary file 5 containing all "n" values and corresponding p-values. In the Figure Legends, we denoted the type of error bars and deviations, included p-values, and referred to the relevant tables for comprehensive numerical data.

Minor comments• On page 8, cell death is visualized using "the apoptotic marker Cas3". It should be Caspase-3. Moreover, it is not clear whether this antibody (directed against vertebrate Caspase-3) recognizes indeed Caspase-3 in *Drosophila*? This should be formulated more carefully.

As the reviewer correctly noted, the Caspase-3 antibody is designed for human Caspase-3. While it has been employed in *Drosophila* apoptosis research, its specificity for Caspase-3 in *Drosophila* is unclear. Given the very well-documented apoptosis in *sws* mutants (Kretzschmar *et al.,* 1997; Muhlig-Versen *et al.,* 2005) and the non-focus on neuronal cell death in this research, we have opted to exclude this information from the supplementary figure. We appreciate the reviewer for bringing this to our attention and for the valuable suggestion.

• On Page 9 (3rd paragraph), the authors report that they "want to understand what signaling pathway is activated." However, the described experiments do not lead to a signaling pathway, but conclude that an antiflammatory response is evoked. This should thus be reworded.

Thank you for pointing this out. In the revised version, we state that we wanted to understand whether the compromised brain barrier in *sws* mutants triggers the activation of any cellular stress pathways, including apoptosis, ferroptosis, oxidative stress, ER stress, and inflammation.

• Figure 1 reports the expression pattern and phenotype of sws; thus, the title of the figure should be extended.

Thank you for the suggestion. We have updated the title of Figure 1 to more accurately reflect its content. The revised title is now: NTE/SWS is expressed in *Drosophila* brain and its loss leads to severe neurodegeneration.

• Concerning the description of phenotypes, the authors use the term "clumps", but it is not clear what this entails (e.g., Page 6, or Figure 6). For the reader, it is also necessary to refer to original studies of moody to understand the septate junction phenotype represented in the figure.

As the reviewer suggested, we changed the word “clumps” to “clusters”. We also agree with the reviewer’s recommendation to cite the original work on Moody to acknowledge previous research and enhance the understanding of *moody* phenotypes. We have now included the relevant citations in the manuscript.

Referees cross-commentingI fully agree with the comments of the other two reviewers, as they were complementary and overlapping with mine (e.g. the contribution of age).Reviewer #1 (Significance (Required)):This study provides a detailed cellular and functional characterization of the swiss cheese phenotype in the blood-brain barrier so far not reported in previous studies, including the team's own earlier publications (e.g., Kretzschmar et al., 1997; Melentev et al., 2021 and Ryabova et al., 2021). Furthermore, it uses cutting-edge technology to provide links to neuroinflammation and neurodegeneration, Previous studies explored neuroinflammation in the brain of *Drosophila* by challenging the organism with bacteria to mount an inflammatory response (Winkler et al., 2021). Intriguingly, this current study provides evidence, that a leaky blood brain barrier alone could lead to an inflammatory response, and that in turn, treatment with anti-inflammatory agents could reduce the cellular defects in glia and in consequence neurodegeneration. This represents an important conceptual advance that will be of wide interest to neurobiologists interested in glial biology, neuroinflammation and neurodegeneration in *Drosophila* and in vertebrates. One possible limitation of the study may be that while complex cellular processes have been pinpointed, some of the causative links of the BBB with neuroinflammation remain unexplored, in particular the aspect of elevated free fatty acids/antimicrobial peptides.

We appreciate the reviewer's recognition of the conceptual significance of our study, revealing that a leaky blood-brain barrier alone can induce an inflammatory response, with subsequent treatment using anti-inflammatory agents and the importance of these findings for neurobiologists. We also thank the reviewer for thorough examination and insightful suggestions. Given that prior studies have demonstrated the induction of neurodegeneration by the overactivation of innate immune-response pathways, especially elevated expression of antimicrobial peptides (Cao *et al.,* 2013), our new experimental data showing increased levels of antimicrobial peptides in aging flies with a defective BBB further strengthen the connection between the BBB, AMPs and neuroinflammation. This link is even more enhanced by the rescue experiments and the detection of macrophage entry in the mutant brains. We trust that the implemented revisions, accompanied by supplementary experimental data, enhance the suitability of our manuscript for publication.

Reviewer #2 (Evidence, reproducibility and clarity (Required)):The manuscript by Tsap et al. describes a role of NTE/SWS in forming the BBB in *Drosophila*. Disruption of the BBB in SWS mutants and knockdown flies results in morphological changes of the glia forming the BBB, increased brain permeability, altered lysosomes, and an upregulation of innate immune genes. The experiments to show a function of SWS in surface glia and the resulting changes in permeability are well supported by the experiments and the statistics appears appropriate.The authors also show changes in innate immune genes and some fatty acids and that similar changes are found in another mutant affecting the BBB. They discuss that these changes are a consequence of the disruptions of the BBB but also that these changes induce changes in the BBB. To address this and confirm that the changes in immune genes and fatty acids is a consequence of the altered BBB, they should include experiment expressing SWS in the surface glia and measure if that normalizes these changes. Another major aspect that should be addressed is the effect of aging. As the authors point out, loss of SWS causes age-dependent phenotypes (shown by the author and others) and with the exception of figure 3F, the age isn't even mentioned in any of the other figures. Furthermore, at least some of the experiments should be done at different ages to determine whether the phenotype is progressive; this includes the permeability assays and the measurements of immune genes (the latter could also support whether changes in the immune genes affect the BBB or vice versa the BBB changes cause the upregulation of immune genes).

As the reviewer suggested, in order to establish a connection between age-dependent correlation between neurodegeneration and inflammation, we analyzed the mRNA expression levels of antimicrobial peptides in the heads of both 15- and 30-day-old *sws* loss-of-function mutants, as well as in flies with *sws* downregulation specifically in SPG cells (*moody>sws^RNAi^*). We found that the expression levels of the antimicrobial peptides are increased in the age-dependent manner in the tested mutants (Figure 5D, red and orange bars). Following the reviewer’s recommendation, we also performed an experiment where we expressed NTE/SWS in the surface glia in a *sws* mutant background (*sws^1^; moody>sws*, rescue). We measured mRNA levels of Attacin A, Cecropin A, and Diptericin in the heads of 15- and 30-day-old flies (Figure 5D, blue bars). The results showed that the levels of all three AMPs were not significantly different or slightly upregulated in the heads of “rescue” animals compared to *Oregon R* controls (Figure 5D, compare green and blue bars, and see Supplementary file 1). Importantly, the levels of all AMPs were significantly lower in the heads of 30-day-old rescue animals than in the heads of the same age *sws^1^* mutants (Figure 5D, compare red and blue bars, green stars, see also Supplementary file 1). These findings further support our hypothesis that *sws* deficit in the surface glia induces an immune response in age-dependent manner. We did not conduct the Dextran permeability assay in older flies because approximately 90% of the 15-day-old flies with *sws* deregulation already exhibited impaired permeability of the BBB. This suggests that the phenotype is quite severe and may not show significant age-dependent progression. Moreover, older mutant flies were extremely weal, and it is likely that they would not have survived the peritoneal injection procedure.

Lastly, the authors claim that septate junctions are defective in sws mutants. However, this should be confirmed by EM studies (which the authors have already done) besides immunohistochemistry which doesn't provide enough resolution.

As the reviewer suggested, for a more detailed detection of septate junctions, we conducted additional electron microscopy experiments. The images included in Figure 6D-F show irregular aggregates and disruptions in the structures of septate junctions and membranes in *sws* mutants compared to controls. Additionally, we display the appearance of tight junctions in *moody* mutants (Figure 5 —figure supplement 2E-F), which look dramatically different compared to *sws* junctions and, as previously described, appear overgrown.

Reviewer #2 (Significance (Required)):A role of SWS in maintaining the BBB and what consequences this has provides another insight how this protein (and its homolog NTE) affects brain health. Although a function of SWS in glia (as well as in neurons) has previously been described, changes in the surface glia and the BBB is a novel aspect. However, the causative role of SWS on some of the described consequences (see above) should be confirmed. Although the manuscript can add to a better understanding of the connection between disruptions of the BBB and neurodegenerative diseases, which is of interest for a broader field of researchers, the discussion of the results is quite speculative.

We appreciate the recognition of the novelty of this work and its potential contribution of our manuscript to a better understanding of the connection between disruptions of the BBB and neurodegenerative diseases. We thank the reviewer for the constructive feedback and hope that introduced changes, along with additional experimental data that address the concerns raised, strengthen the proposed role of *sws* in the formation of tight junctions in the BBB and its age-dependent maintenance.

Reviewer #3 (Evidence, reproducibility and clarity (Required)):Summary:The study of the formation and maintenance of the blood-brain barrier (BBB) is a growing field of study, partly due to its strong link with neurological disorders. The BBB depends on the role of multiple cell types and mechanisms. Mutations in the conserved phospholipase NTE/SWS can lead to neurodegeneration, and previous work from the authors shows that SWS loss leads to abnormal glial morphology. In this work, authors use *Drosophila* to further study this phenotype, showing that SWS is mostly expressed in the BBB-related glia and that its loss leads to abnormal BBB permeability, increased inflammatory response and neural cell death. Interestingly, authors observed a dependence for the BBB-defective phenotype on aging, with important implications for SWS/NTE and neurodegeneration. Overall, the work represents a clear advance in the poorly explored role of NTE/SWS in neurodegeneration, with a broad impact on the understanding of BBB maintenance. This work shows a combination of multiple and appropriate experimental approaches, including confocal microscopy, EM, RT-qPCR, or gas chromatography-mass spectrometry among others.Major comments:The use of sws1 and sws1/sws4 transheterozygous animals, together with the use of sws RNAi is a solid approach to validate that the reported phenotypes are due to SWS loss. Using these models, the authors performed a convincing structural analysis of the subperineurial glia phenotype, and showed that it is accompanied by a defective BBB, inflammation and neuronal cell death. The key conclusions are properly supported by the data. However, there are some claims in the text that are not supported by any data in the Figures, but only qualifications. This needs to be fixed:-Page 6, third paragraph:"…we specifically downregulated sws in the nervous system using the double driver line that allows downregulation of sws in glia and neurons (repo, nsyb-Gal4, Suppl. Figure 2C-Cʹ). Since these animals had the same disorganized structure of brain surface as the loss-of-function mutant…"Supp. Figure 2C-C' only shows expression of CD8:GFP and nlacZ reporters by repo and nsyb-Gal4, but there is no data showing sws RNAi expression by these drivers.

We thank the reviewer for noticing these referencing mistakes. We have corrected the references to the expression patterns of the glial and/or neuronal *Gal4* drivers (Figure 2 —figure supplement 1D, E and F). Bar graph in Figure 2 —figure supplement 1C shows RT-qPCR analysis of *sws* mRNA levels from flies with glial and/or neuronal *sws* downregulation (repo*>sws^RNAi^, nsyb>sws^RNAi^* and *repo, nsyb>sws^RNAi^*), and the images of mutant brains in Figure 2 —figure supplement 2 and Figure 2A-C show the surface glia phenotypes in these mutants.

"…Moreover, downregulation of sws in all glial cells (repo>swsRNAi) resulted in the same phenotype. At the same time, upon sws downregulation in neurons,… (Suppl. Figure 4)…"Suppl. Figure 4 only shows nsyb>swsRNAi data but not repo>swsRNAi

We show now both *repo>sws^RNAi^* and *nsyb>sws^RNAi^* (Figure 2 —figure supplement 2C and 2E, respectively).

-Page 6, fourth paragraph:"Importantly, expression of *Drosophila* or human NTE in these glia cells rescued this phenotype (Figure 2H)"In addition to the indicated quantifications, it is essential to show some representative data showing the phenotype when *Drosophila* or human NTE are expressed in glial cells of sws mutant animals.

We agree with the reviewer that it is important to show the rescue phenotypes. We have included images of the brain surface of *sws* mutants that have *Drosophila* or human NTE expressed in glial cells (Figure 2D and Figure 2 —figure supplement 2F).

-Page 9, last paragraph: "We found that in moody mutants, the surface glia phenotype analyzed using CoraC as a marker could also be suppressed by NSAID and rapamycin (Figure 5A)."In addition to the indicated quantifications, it is essential to show some representative data showing the phenotype with and without treatments.

We appreciate the reviewer's suggestion, and as recommended, we have included representative data showing the phenotype with and without treatments in Figure 5 —figure supplement 1C-D.

A more detailed analysis of two aspects of the data would clearly improve the manuscript, whose findings are a bit superficial in the current state:- The exact mechanism by which BBB permeability leads to brain inflammation remains unknown. Authors show that accumulation of polyunsaturated fatty acids (known to regulate inflammation) occurs in sws-depleted animals. However, they only observed a correlation between this phenotype and the inflammatory response, while is not clear whether the accumulation of polyunsaturated fatty acids causes inflammation in this model or is a consequence of it. An attempt to rescue the accumulation of polyunsaturated fatty acids (i.e., knocking down a required enzyme for their production) in sws mutants might help to understand this. Also, the fact that the defective BBB phenotype observed in either sws KO and glia-specific KD can only be partially rescued by the use of inflammation inhibitors, suggests that other pathways are involved.

We agree with reviewer that since the use of inflammation inhibitors only partially rescue the defective BBB phenotype in *sws* mutants, it implies the involvement of additional pathways. While our data reveal a correlation between the accumulation of polyunsaturated fatty acids and the inflammatory response, whether this accumulation causes inflammation in our system remains to be studied. We have revised the text to ensure that this explanation is clearly stated without overemphasis.

- While the differences between the phenotypes caused by sws or moody loss are well characterized, it would be key for this work to further study the mechanisms by which sws controls septate junctions. The authors propose the organization of lipid rafts, but some experiments in that direction to check this hypothesis. For example, can authors reproduce the septate junction phenotype of sws mutant (Figure 6C) by using a different approach to induce defective lysosomes in subperineurial glia?

We appreciate the reviewer's suggestion for such an insightful experiment. To investigate whether the septate junction phenotype observed in *sws* mutants can be replicated in mutants with defective lysosomes in subperineurial glia, we downregulated several key lysosomal genes in SPG cells: *moody>Dysb^RNAi^, moody>Npc1a^RNAi^, moody>Pldn^RNAi^,* and *moody>spin^RNAi^* (Figure 6 —figure supplement 1A-E). We were happy to see that downregulation of any of these genes resulted in abnormal formation of SJs and membrane organization in SPG cells. These additional experiments strongly support our hypothesis that lysosomal control of membrane homeostasis significantly impacts the appearance of SJs. Thank you for this excellent idea.

The attempt of the proposed approaches above should require about 3-6 months of investment, with limited economic effort, given the availability and diversity of lines found in the existing stock centres such as Bloomington or Vienna.The data is presented very clearly, and the methods are adequately detailed, and the experiments and statistical analysis are adequate.Minor comments:Prior studies are referenced appropriately, but there is a case that should be addressed. On Page 3, first paragraph, regarding the sentence: "However, the molecular mechanisms underlying inflammaging remain unclear". I recommend specifying what is known and what is unknown in the field. Ideally describing (briefly) the knowledge about lipids, inflammaging and neurodegeneration, which are the specific topics of the research. Otherwise, the current sentence is too vague, while there is a lot of work published about it.

As the reviewer suggested we have extended the first part of our introduction to briefly describe how inflammaging is connected with the BBB, fatty acid metabolism and lysosomal functions.

The text and figures are clear and accurate. The logic of the experiments and the results are exposed very clearly (for example, the Suppl. Tables are very helpful). There are a few minor issues, however, that should be addressed:- Page 4, first paragraph: regarding the sentence: "For various obvious reasons, humans are not ideal subjects for age-related research.", I recommend specifying the main reasons (i.e. life cycle, ethical issues, etc.?).

Thank you, the main reasons are specified now.

- I would recommend moving the text "For various obvious reasons…disrupted upon ageing." From its current position to just before "*Drosophila melanogaster* is an excellent…". This would keep a better logic in the text by explaining NTE first and later introducing the models to study its function. Presenting then *Drosophila*.

Thank you, done.

- To support the sentence "Together, *Drosophila* satisfies…neurodegeneration during aging", instead of citing so many papers, I recommend citing just one current review about it, since the amount of literature supporting the claim is huge and should not be limited to a few "random" articles. An alternative might be indicating that the lab has used *Drosophila* for this aim before, and then citing the examples from the literature.

Thank you for this suggestion, now we referenced few recent reviews and referred to our previous work on the topic.

- Page 4, second paragraph: if NTE/SWS is going to be used as a synonym for NTE/SWS loss of function (or other type) model, it needs to be specified. Otherwise, refers to the proteins and sentences like "NTE/SWS has been shown to result in lipid droplet accumulation…" are misleading.

Thank you for the suggestion; we have now specified that NTE/SWS is used as a synonym for the SWS protein in *Drosophila* and corrected this throughout the manuscript.

- Page 4, last paragraph: the first time that "BBB" is used, its meaning should be specified. And three lines below use "BBB" instead of blood-brain barrier.

Thank you, corrected.

Referees cross-commentingI agree with the comments provided by the other reviewers. They are well reasoned and cover some aspects of the work that I did not see. Regarding the main issue, the three revisions point at the same direction, that is the limited analysis about the mechanism underlying the phenotypes.Reviewer #3 (Significance (Required)):- Describe the nature and significance of the advance (e.g. conceptual, technical, clinical) for the field.This work represents a substantial advance in the understanding of NTE/SWS function in the context of neurodegeneration, and opens potential approaches to treat related disorders (they successfully use anti-inflammatory compounds to ameliorate some of the key phenotypes). However, the findings are a bit superficial in terms of mechanisms, and further analysis (see major comments) would notably improve the significance of the manuscript. This should be realistic and suitable, given the advantages of the *Drosophila* model and the availability of tools.- Place the work in the context of the existing literature.The role of SWS in regulating lysosomal function is potentially supported by NTE-deficient mice data (Akassoglou et al., 2004; Read et al., 2009), where different types of neurons show similar dense bodies containing concentrically laminated and multilayered membranes than those observed in this work in *Drosophila* sws mutant. Potentially, the rest of the work has a translation to mammals, which is supported by the fact that ectopic expression of NTE rescues some of the key phenotypes described in the manuscript.- State what audience might be interested in and influenced by the reported findings.Neuroscience in general, since the study of BBB and neurodegeneration has a clear general interest in the whole field.- Define your field of expertise with a few keywords to help the authors contextualize your point of view. Indicate if there are any parts of the paper that you do not have sufficient expertise to evaluate.*Drosophila*; Neurodegeneration; Hereditary Spastic Paraplegia; Alzheimer's disease; Motor neurons; Microglia; Endoplasmic reticulum; Mitochondria.Lipid metabolism is the part of the manuscript where I have less expertise to evaluate, only having general knowledge about it.

We appreciate the positive evaluation of our work, the careful reading, and the valuable suggestions provided by the reviewer, including recommendations for additional experiments and changes in the text. We believe that the implemented changes, combined with the new experimental data, have improved the manuscript, making it ready for publication.

References

Artiushin G, Zhang SL, Tricoire H, Sehgal A (2018) Endocytosis at the *Drosophila* blood-brain barrier as a function for sleep. *ELife* 7

Cao Y, Chtarbanova S, Petersen AJ, Ganetzky B (2013) Dnr1 mutations cause neurodegeneration in *Drosophila* by activating the innate immune response in the brain. *Proc Natl Acad Sci U S A* 110: E1752-1760

Kretzschmar D, Hasan G, Sharma S, Heisenberg M, Benzer S (1997) The swiss cheese mutant causes glial hyperwrapping and brain degeneration in *Drosophila*. *J Neurosci* 17: 7425-7432

Muhlig-Versen M, da Cruz AB, Tschape JA, Moser M, Buttner R, Athenstaedt K, Glynn P, Kretzschmar D (2005) Loss of Swiss cheese/neuropathy target esterase activity causes disruption of phosphatidylcholine homeostasis and neuronal and glial death in adult *Drosophila*. *J Neurosci* 25: 2865-2873